# Seesaw: Accelerating Training by Balancing Learning Rate and Batch Size Scheduling

**Alexandru Meterez**[*]
Harvard University
ameterez@g.harvard.edu

**Depen Morwani**[*]
Harvard University
dmorwani@g.harvard.edu

**Jingfeng Wu**
University of California, Berkeley
uuujf@berkeley.edu

**Costin-Andrei Oncescu**
Harvard University
concescu@g.harvard.edu

**Cengiz Pehlevan**
Harvard University
cpehlevan@seas.harvard.edu

**Sham Kakade**
Harvard University
sham@seas.harvard.edu

## Abstract

Increasing the batch size during training — a "batch ramp" — is a promising strategy to accelerate large language model pretraining. While for SGD, doubling the batch size can be equivalent to halving the learning rate, the optimal strategy for adaptive optimizers like Adam is less clear. As a result, any batch-ramp scheduling, if used at all, is typically tuned heuristically.

This work develops a principled framework for batch-size scheduling and introduces *Seesaw*: whenever a standard scheduler would halve the learning rate, Seesaw instead multiplies it by $1/\sqrt{2}$ and doubles the batch size, preserving loss dynamics while reducing serial steps. Theoretically, we provide, to our knowledge, the first finite-sample proof of equivalence between learning-rate decay and batch-size ramp-up for SGD on noisy linear regression, and we extend this equivalence to normalized SGD, a tractable proxy for Adam, under a variance-dominated regime observed in practice. Empirically, on 150M/300M/600M-parameter models trained at Chinchilla scale using a constant (critical) batch size, *Seesaw* matches cosine decay at equal FLOPs while reducing wall-clock time by $\approx 36\%$, *approaching the theoretical limit* implied by our analysis.

## 1 Introduction

In recent years, large language models (LLMs) have demonstrated remarkable progress across diverse tasks, including outperforming humans in competitive benchmarks and international competitions (Huang & Yang, 2025; Petrov et al., 2025; El-Kishky et al., 2025). A central driver of this progress has been the steady increase in pre-training compute, measured in floating point operations (FLOPs) (Kaplan et al., 2020; Hoffmann et al., 2022). However, hardware improvements have not kept pace with the rapid escalation of training requirements, resulting in wall-clock times extending to several months for state-of-the-art models (Erdil & Schneider-Joseph, 2024).

A widely studied strategy to reduce wall clock time is increasing the batch size (You et al., 2017; Goyal et al., 2017). Empirical studies show that larger batches can proportionally reduce the number of optimization steps required for convergence (Zhang et al., 2024; McCandlish et al., 2018; Shallue et al., 2019). However, beyond a maximum batch size termed as critical batch size (CBS), further scaling reduces sample efficiency and limits gains in training speed.

While most prior work assumes a fixed batch size, recent large-scale LLM training runs employ batch size schedules that gradually increase batch size over the course of training (Dubey et al.,

---

[*]Equal contribution

2024; Touvron et al., 2023; Adler et al., 2024; OLMo et al., 2024; Team, 2025). This practice has been observed to further reduce training times without compromising model performance. However, to the best of our knowledge, the "batch ramp" schedules are not theoretically grounded and instead tuned heuristically. The lack of theoretical justification leaves open whether these heuristics are close to optimal, motivating the central question of our study: *what is the optimal batch size schedule for minimizing serial runtime while not sacrificing performance?*

## 1.1 THEORETICAL CONTRIBUTIONS

We theoretically prove, to the best our knowledge, the first non-asymptotic equivalence result between learning rate decay and batch size ramp up in SGD in linear regression with additive noise. We introduce an informal version of our main theorem here, as well as the corollary leading up to Seesaw, and we formalize the statements in Section 5.

**Theorem** (Informal version of Theorem 1). *Consider mini-batch SGD on D total samples. Consider a base process where we run with a stepwise batch ramp up schedule which doubles the batch size at certain points while keeping the learning rate fixed. Consider an alternative process where at the same points we instead halve the learning rate, while keeping the batch size fixed and adjust the number of steps such that the total processed samples remains D. Then, the excess risk of the base process is within a constant factor of that of the alternative process.*

**Corollary** (Informal version of Corollary 1). *Under mild assumptions, we extend the equivalence to normalized SGD with different schedulers. Consider a base process where we run with a stepwise batch ramp up schedule which doubles the batch size at certain points while decaying the learning rate by $\sqrt{2}$. Consider the same alternative process as before. Then, the excess risk of the base process is within a constant factor of that of the alternative process.*

## 1.2 EMPIRICAL CONTRIBUTIONS

Based on the theoretical analysis, we introduce *Seesaw*, a learning rate and batch size scheduler that reduces the serial runtime of LLM pre-training runs by approximately $36\%$ via increasing the batch size during training at specific points. We provide empirical results in Figure 1 and show that at (or below) the critical batch size, our method achieves a significant serial runtime acceleration across several model and data scales, while maintaining the same performance as training with cosine decay.

## 2 RELATED WORK

**Role of batch size in scaling.** Understanding batch size ramp up schemes during training has been a topic of interest in recent years due to its crucial role in decreasing wall clock runtime. Various methods of increasing the batch size have been used in common LLMs such as LLaMA (Dubey et al., 2024; Touvron et al., 2023), Nemotron (Adler et al., 2024), OLMo (OLMo et al., 2024; Groeneveld et al., 2024), Apertus (Team, 2025). The reason behind ramping up the batch size is to take advantage of the parallel computation of samples and thus reducing the total number of sequential steps. However, since increasing the batch size reduces the total number of gradient steps taken by the model during training, there is a maximal batch size which can be achieved without becoming data inefficient, called the critical batch size (CBS) (Erdil & Schneider-Joseph, 2024; Jain et al., 2018; Zhang et al., 2024; Shallue et al., 2019). Recent work also looks at the effect of batch size on SGD optimization in LLMs (Srećković et al., 2025; Marek et al., 2025), following previously established theoretical results in noisy quadratic models (Zhang et al., 2019).

**SGD for linear regression.** Recently, Zhang et al. (2024) have analyzed the CBS using weight averaging in linear regression and established scaling laws as a function of data and model size. The bias-variance analysis used by Zhang et al. (2024) has a longstanding history in the literature (Jain et al., 2017) and has been used to study batch ramp-up schemes in SGD (Jain et al., 2018). These rates have been recently made tight by (Zou et al., 2021; Wu et al., 2022a;b) for general spectra of the data covariance. Recently, (Meterez et al., 2025) have used a simplified mathematical framework for rederiving the same bounds by rotating the dynamics in the eigenbasis of the data. A similar

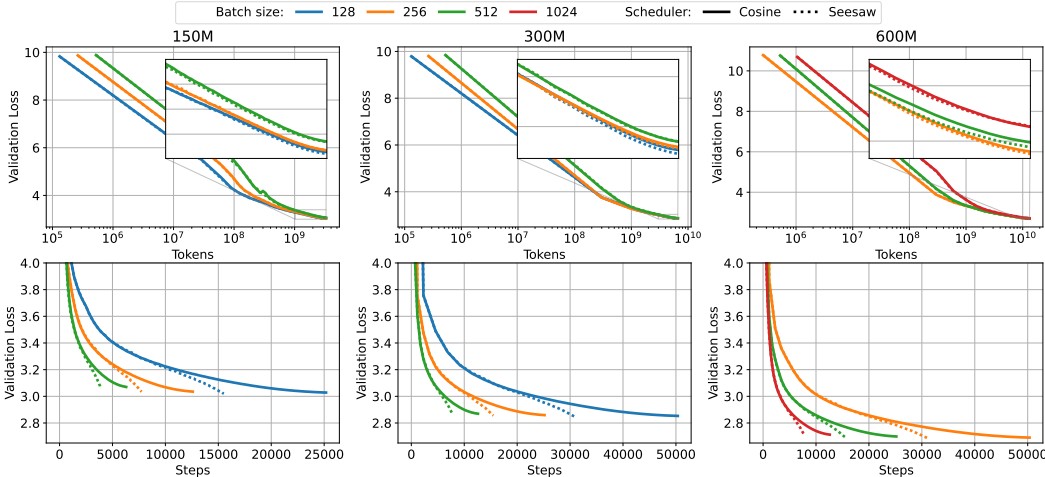

Figure 1: Seesaw comparison with cosine decay in 150M (left), 300M (middle) and 600M (right) models trained at Chinchilla scale. Seesaw matches the loss dynamics of cosine annealing in FLOPs (top row), but achieves a significant speed up in terms of serial runtime (bottom row). Runs are swept over learning rates and plotted at the best learning rate for cosine annealing in terms of validation loss, at each batch size. The validation losses at the end of training are provided in Table 1. Note the axes: the top plots are on a logarithmic scale while the bottom are on a linear scale. For more experimental details, see Section 4.

diagonalizing idea has also been previously used in literature by Bordelon & Pehlevan (2021); Wu et al. (2023b;a).

**Stochastic Differential Equations (SDEs).**     Another point of view for studying the interaction between batch size and learning rate in optimization is through SDEs (Li et al., 2021; Xie et al., 2020; Compagnoni et al., 2024; Jastrzębski et al., 2017). Malladi et al. (2022) study how to scale the learning rate as a function of the batch size in adaptive algorithms, extending previous work that introduced the square root scaling rule (Granziol et al., 2022; You et al., 2019).

**Empirical Work.**    Scaling laws for the CBS and the optimal batch size have also been recently observed by (Bergsma et al., 2025). In line with our conclusions regarding SGD, the linear scaling rule for SGD has been observed by (Smith et al., 2017), showing that in SGD, linearly increasing the batch size is equivalent to decreasing the learning rate. McCandlish et al. (2018) propose a metric based on the Hessian and the noise that correlates with the CBS over training. While their proposed metric is based on having access to the Hessian, which is prohibitive for current large-scale runs, they find that the noise scale increases during a training run, which aligns with our theoretical predictions. Lastly, perhaps the most similar to our work is Merrill et al. (2025), who propose a batch size warmup scheme based on starting from a checkpoint with various multiples $k$ of the current batch size, and pick the largest $k^\star$ where the loss is $\epsilon$-close to the original loss. Based on this methodology, they propose the scaling rule $B_{t+1} = 2B_t$ and $\eta_{t+1} = \sqrt{2}\eta$. In contrast, we propose a simple drop-in replacement for existing cosine schedulers, motivated rigorously by (normalized) SGD on quadratics. Moreover, we argue that the scheduler proposed by (Merrill et al., 2025) will lead to instabilities and divergence after a fixed number of steps, based on our theoretical analysis in Lemma 4.

## 3    SEESAW: ALGORITHMIC DETAILS

We begin by providing an intuitive derivation of Seesaw, and the practical implementation of our algorithm. To build intuition, consider 2 different SGD processes. In one process we take 2 steps at learning rate $\eta/2$ and batch size $B$, and in the other we take 1 step at learning rate $\eta$ and batch size $2B$. Intuitively, both processes should look the same up to first order: the deterministic part of the

update stays the same, and the noise averages out. Consider a general smooth loss function $\mathcal{L}(\mathbf{x})$ and let $\mathbf{g}_0 = \nabla \mathcal{L}(\mathbf{x}_0)$. Then, through a simple Taylor expansion up to first order in $\eta$, we have the loss of the $(\eta, 2B)$ process and the loss of the 2 half step process $(\eta/2, B)$ respectively:

$$\mathcal{L}(\mathbf{x}_1) = \mathcal{L}(\mathbf{x}_0) - \eta \mathbf{g}_0^\top (\mathbf{g}_0 + \xi') + \mathcal{O}(\eta^2) \qquad \text{Cov}(\xi') = \frac{\sigma^2}{2B} \mathbf{I}_d$$

$$\mathcal{L}(\mathbf{x}_2) = \mathcal{L}(\mathbf{x}_0) - \frac{\eta}{2} \mathbf{g}_0^\top (2\mathbf{g}_0 + \xi_0 + \xi_1) + \mathcal{O}(\eta^2) \qquad \text{Cov}(\xi_i) = \frac{\sigma^2}{B} \mathbf{I}_d.$$

Note that the 2 processes are equivalent up to first order both in the deterministic part and in the noise terms up to $\mathcal{O}(\eta^2)$, an argument which has been previously shown by Malladi et al. (2022). We formalize this SGD intuition in Theorem 1 and extend it to normalized SGD as an analytical proxy to Adam.

## 3.1 EXTENSION TO NORMALIZED SGD

From the previous subsection, intuitively, for SGD, cutting the learning rate by a factor of $\alpha$ should be equivalent to increasing the batch size by a factor of $\alpha$. To design a practical training algorithm based on the SGD analysis and arrive at Seesaw, we begin with the Adam update rule and simplify until we obtain normalized SGD (NSGD), which is a commonly used tractable analytical proxy for Adam (Jelassi et al., 2022; Zhao et al., 2024; Xie et al., 2024). Suppose we are optimizing over parameters $\theta$ and denote the gradients at each time step $\mathbf{g}_t$. Then, for learning rate $\eta$ and ignoring the bias correction, the parameter update is:

$$\mathbf{m}_t = \beta_1 \mathbf{m}_{t-1} + (1 - \beta_1) \mathbf{g}_t \tag{1}$$

$$\mathbf{v}_t = \beta_2 \mathbf{v}_{t-1} + (1 - \beta_2) \mathbf{g}_t^2 \tag{2}$$

$$\theta_t = \theta_t - \eta \frac{\mathbf{m}_t}{\sqrt{\mathbf{v}_t} + \epsilon} \tag{3}$$

where $\mathbf{m}_t$ is the momentum term, $\mathbf{v}_t$ is the second moment term, $\beta_1, \beta_2$ are their respective exponential decay rates, and $\epsilon$ ensures stability. For NSGD, we approximate the per-coordinate updates of Adam will full parameter updates, set $\beta_1 = \beta_2 = 0$ and replace the denominator with the true expected value of the squared gradient norms over the population:

$$\theta_t = \theta_t - \eta \frac{\mathbf{g}_t}{\sqrt{\mathbb{E}\|\mathbf{g}_t\|^2}} \tag{4}$$

Equation 4 describes the NSGD update rule, which is a crucial component of designing Seesaw. While the full analysis is deferred to Appendix B, the expected gradient norms can be decomposed as:

$$\mathbb{E}\|\mathbf{g}_t\|^2 = \texttt{mean} + \texttt{variance} \tag{5}$$

where the variance scales down with the batch size. To design Seesaw, we assume that the variance dominates the expected gradient squared norms (Assumption 3), and we motivate why this assumption is reasonable in Appendix B. This step reduces (up to constant factors) the NSGD

---

**Algorithm 1:** Seesaw

**Inputs:** $\eta_0$ (initial learning rate), $B_0$ (initial batch size), $\alpha > 1$ (step decay factor), $S$ (steps at which input scheduler cuts $\eta$ by $\alpha$)
*// For a given input scheduler, $S_i$ are the steps where $\eta = \eta_0/\alpha^i$*
$\eta \leftarrow \eta_0, B \leftarrow B_0, \texttt{scheduler} \leftarrow [\,]$
**for** $t \in S$ **do**
$\quad \eta \leftarrow \eta/\sqrt{\alpha}$
$\quad B \leftarrow B \cdot \alpha$
$\quad \texttt{scheduler.append}(t, \eta, B)$
**end**
**return** scheduler

---

update rule to SGD with a rescaled learning rate, allowing us to extend risk equivalence to NSGD (Corollary 1) in Section 5. For NSGD, informally, Corollary 1 shows that any learning rate cut by a factor of $\alpha$ and batch size increase by a factor of $\beta$ are equivalent as long as $\alpha\sqrt{\beta}$ is held constant. We further empirically compare Seesaw with other possible schedulers in Figure 4.

## 3.2 ACHIEVABLE SPEEDUPS

While our theory is established for step decay schedulers, in practice we approximate cosine decay with a step decay by considering a decay of $\alpha$, and passing the times (as measured in tokens) where

the cosine would cut the learning rate by $\alpha$ as input to Seesaw. Then, at these points, we instead cut the learning rate by $\sqrt{\alpha}$ and increase the batch size by $\beta$, where the schedulers are equivalent in terms of loss as long as we keep the product $\alpha\sqrt{\beta}$ fixed. However, we cannot arbitrarily increase the batch size at time $t$ and expect the risk to match the underlying process. Lemma 4 quantifies this and the main takeaway is stated below:

**Remark 1.** *The most aggressive ramp up scheme we can use is given by $\alpha = \sqrt{\beta}$. (for a formal argument see Lemma 4)*

In Section 4.1 we empirically verify this constraint and show that $\alpha = \sqrt{\beta}$ is the most aggressive scheme we can choose without divergence, which is the reason for presenting Algorithm 1 in this setting.

At the most aggressive limit, we can compute the theoretical speedup we would hope to achieve where the standard scheduler is the cosine decay.

**Lemma 1** (Maximum Theoretical Speedup under Cosine Decay). *Consider a baseline training process of $T$ total steps using a constant batch size and a cosine learning rate schedule $\eta(t) = \eta_0 \cos(\frac{\pi t}{2T})$. An equivalent process run with a batch ramping schedule like Seesaw, in the continuous limit [1], will have a total of $\frac{2T}{\pi}$ steps. This yields a maximum theoretical serial runtime reduction of $(1 - \frac{2}{\pi}) \approx 36.3\%$.*

Lemma 1 provides an intuitive upper bound on the acceleration from *Seesaw*. The speedup is significant but less than 50% because most of the training progress under a cosine schedule occurs early, when the learning rate is high and the batch size must consequently be relatively small. While *Seesaw* aggressively increases parallelism in the later stages of training, the initial, more sequential phase remains the primary bottleneck on total runtime.

## 4 EMPIRICAL FINDINGS

In this section, we present the experimental details and methodology for evaluating Seesaw. We denote by $D$ the dataset size, $N$ the number of parameters.

|  | B=128 | B=256 | B=512 | B=1024 |
|---|---|---|---|---|
| 150M (cosine) | 3.0282 | 3.0353 | 3.0696 | 3.1214 |
| 150M (Seesaw) | 3.0208 | 3.0346 | 3.0687 | 3.1318 |
| 300M (cosine) | 2.8531 | 2.8591 | 2.8696 | 2.9369 |
| 300M (Seesaw) | 2.8452 | 2.8561 | 2.8700 | 2.9490 |
| 600M (cosine) | - | 2.6904 | 2.6988 | 2.7128 |
| 600M (Seesaw) | - | 2.6883 | 2.6944 | 2.7132 |

Table 1: Final validation losses picked at the best learning rate (for the cosine annealing scheduler) for each batch size, for $\alpha = 1.1$. Note that the dynamics match robustly across the 2 schedulers when trained at CBS.

**Model and Dataset.** We pretrain models of size 150M, 300M and 600M (non-embedding) parameters at Chinchilla scaling i.e. $D = 20N$ (Hoffmann et al., 2022). We use the OLMo (Groeneveld et al., 2024) codebase to train all of our models. For each experiment, we do learning rate warmup for 10% of the total amount of tokens, followed by learning rate decay following cosine scheduling or Seesaw. We report the architectural details of each model as a tuple (depth, # heads, width), and thus we have for 150M $(12, 16, 1024)$, 300M $(24, 16, 1024)$ and for 600M $(24, 22, 1408)$. Unless mentioned otherwise, each model is trained using AdamW, with weight decay $\lambda = 0.0$ (no weight decay), $\beta_1 = 0.9$, $\beta_2 = 0.95$, $\epsilon = 10^{-8}$. For each run we sweep over learning rates $\eta \in \{0.001, 0.003, 0.01, 0.03\}$ and initial batch sizes $B \in \{128, 256, 512, 1024\}$, at sequence length

---

[1] In the continuous-time limit, we consider an aggressive (non-divergent) batch size ramp that maintains the relationship $\alpha = \sqrt{\beta}$. Consequently, the total number of sequential steps is given by the integral of the normalized learning rate schedule: $\int_0^T \frac{\eta(t)}{\eta_0} dt = \int_0^T \cos(\frac{\pi t}{2T}) dt = \frac{2T}{\pi}$.

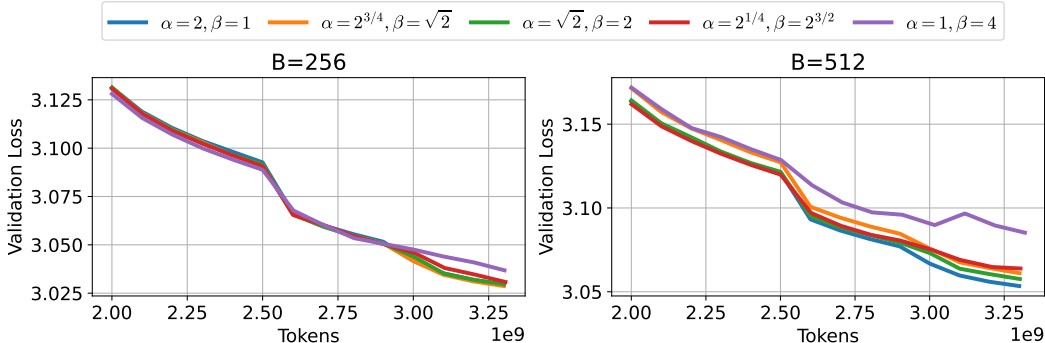

Figure 2: 150M models trained at batch size 256 (left) and 512 (right) with $\alpha$ and $\beta$ values following the line of equivalence $\alpha\sqrt{\beta} = 2$ described in Table 2. Note that the target to match is the blue trace, and our theory (Lemma 4) predicts that the red and purple traces should not match the baseline (blue trace) due to instabilities.

$L = 1024$. Similar to the OLMo training codebase, we enable z-loss during training, but provide ablations over it in Appendix D showing that it does not affect the model performance at our scales. All our models are pretrained on the C4 dataset (Raffel et al., 2020), tokenized with the T5 tokenizer.

**Experimental Design.** We compare Seesaw with cosine annealing by training models at the critical batch size (CBS) $B^\star$, approximated based on (Zhang et al., 2024), namely $B^\star \approx 256 \times L$ (150M), $B^\star \approx 512 \times L$ (300M) and $B^\star \approx 1024 \times L$ (600M) tokens. The main results comparing Seesaw and cosine annealing at equal FLOPs are provided in Figure 1. The precise final losses obtained by the 2 schedulers are provided in Table 1.

## 4.1 CAN WE DO BETTER?

Recall that based on Corollary 1 and Lemma 4, we have a family of equivalent schedules in NSGD, given by a fixed product $\alpha\sqrt{\beta}$, under the constraint that $\alpha \geq \sqrt{\beta}$. Ideally, we would like to make $\beta$ as large as possible, since this would lead to larger batch sizes, and thus assuming enough devices are available, the lowest serial runtime. Crucially, the constraint prevents us from using a too agressive batch size scheduler. In this section, we empirically verify our theoretical prediction by testing schedulers positioned at various points on the $(\alpha, \beta)$ axis. Namely, we train 150M models at fixed

| $\alpha$ | 2 | $2^{1/4}$ | $2^{1/2}$ | $2^{3/4}$ | 1 |
|---|---|---|---|---|---|
| $\beta$ | 1 | $2^{3/2}$ | 2 | $2^{1/2}$ | $2^2$ |

Table 2: $\alpha, \beta$ values used to test the extreme values of the equivalence.

batch size and Chinchilla scale, and we approximate cosine decay with a step decay scheduler that halves the learning rate at the token counts where the cosine schedule's learning rate would halve. This gives us the baseline $\alpha = 2$ and $\beta = 1$, with the product $\alpha\sqrt{\beta} = 2$. Based on the theoretical constraint and the equivalence line, the most aggressive scheduler we could use is $\alpha = \sqrt{2}$ and $\beta = 2$. To validate our hypothesis, we compare with $\alpha = 1$ and $\beta = 4$, and points in between at geometric intervals. Table 2 gives an overview of the experimental design, and Figure 2 shows that indeed the most aggressive schedules tend to underperform.

## 4.2 WHEN DOES ASSUMPTION 3 FAIL?

Up to this point, a crucial assumption for the development of our theory and the design of Seesaw has been Assumption 3. Recall that Assumption 3 states that the expected gradient norms – namely, the denominator of the NSGD update step, is dominated by the additive noise. Intuitively, since the noise variance decreases with the batch size as $\mathcal{O}(1/B)$, one can see that past a certain batch the additive noise will become small, and thus Assumption 3 will fail. In Figure 3, we can see that at sufficiently

large batch sizes, indeed Seesaw starts to perform worse as compared to the underlying cosine schedule. The first hypothesis could be that it is still possible to match the underlying schedule, but

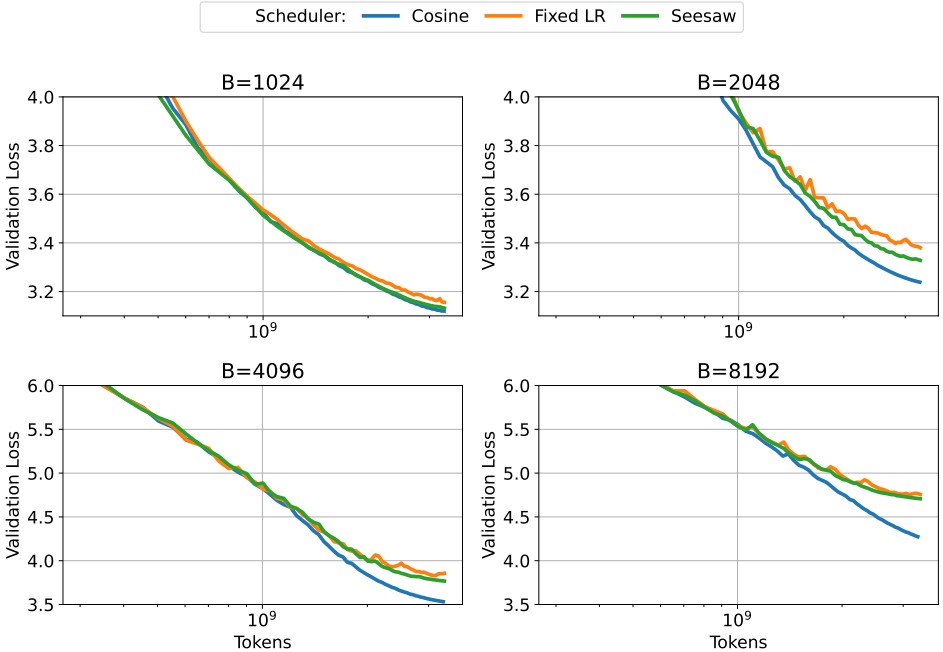

Figure 3: 150M models trained past CBS (roughly 256), at batch sizes 1024, 2048, 4096 and 8192, for 3 schedulers: cosine decay (blue), constant learning rate with increasing batch size based on Seesaw (orange) and Seesaw (green). Note that none of the proposed schedules is able to match the cosine curve, with the discrepancy increasing as the batch size grows more.

with a learning rate equivalence as given by `mean` dominating in the denominator. As `mean` does not scale with batch size, therefore, using the equivalence schedule as required by SGD could be a promising candidate. We explore this option in Figure 3, and it turns out that this schedule performs even worse than the Seesaw schedule. We hypothesise that beyond a certain batch size, it is not possible to match the performance of learning rate decay by any equivalent batch size ramp up for Adam or normalized SGD, which we motivate using the following toy example.

For simplicity, we look at NGD in 1D, for the quadratic loss $\mathcal{L}(x) = \frac{1}{2}hx^2$, where $x, h \in \mathbb{R}$ and $h \geq 0$. Training with NGD, we have the loss gradients with respect to the parameters and the update rule:

$$\nabla_x \mathcal{L} = hx \qquad x_{t+1} := x_t + \eta h \operatorname{sign}(x_t) \qquad \Delta_t = \eta h \operatorname{sign}(x_t)$$

where $\operatorname{sign}(x_t) = \frac{x_t}{|x_t|}$, and $\Delta_t = x_t - x^\star$ is the distance of the current iterate from the minimizer. Note that if $x_t > 0$, then $\Delta_t = \eta h$ and if $x_t < 0$, then $\Delta_t = -\eta h$, implying that the model does not reach the minimizer and instead converges to a stable cycle of $\mathcal{O}(\eta)$ around the minimizer. In order to escape this stable cycle and reach the minimizer, it is thus necessary to decay the learning rate. Therefore, if we slightly relax the setup and think of a large training batch as being close to NGD regime, we can see that further increasing the batch size does not change the dynamics. Therefore, past a certain batch size, it is fundamentally impossible to formulate a batch size ramp up scheme with fixed learning rate that achieves the same loss as a learning rate scheduler at fixed batch size.

## 5 THEORETICAL ANALYSIS

In this section we introduce the main theoretical contributions of our work. Namely, under mild assumptions, we establish a formal equivalence between learning rate decay and batch size ramp up in SGD and normalized SGD.

**Setup and Notation.** We use the notation $f \lesssim g$ to mean that there exists some constant $c > 0$ such that $f(x) \leq cg(x)$ for any $x$. We also use the notation $f \asymp g$ if $f(x) \lesssim g(x) \lesssim f(x)$ for all $x$. We denote the samples $(\mathbf{x}, y)$ where $\mathbf{x} \in \mathbb{R}^d$ and $y \in \mathbb{R}$, with the distribution and risk:

$$\mathbf{x} \sim \mathcal{N}(0, \mathbf{H}) \qquad y|\mathbf{x} \sim \mathcal{N}(\langle \mathbf{w}^\star, \mathbf{x} \rangle, \sigma^2) \qquad \mathcal{R}(\mathbf{w}) = \frac{1}{2}\mathbb{E}(\langle \mathbf{w}, \mathbf{x} \rangle - y)^2$$

where the expectation is over the $(\mathbf{x}, y)$, $\mathbf{w}^\star$ is the minimizer, and $\sigma^2$ is the variance of the additive noise. We also use $\mathcal{R}(\mathbf{w}_t, \eta)$ to denote the risk at time $t$ for a process trained with $\eta$, but we drop the $\eta$ parameter when it is clear from context. We consider step decay schedules for the learning rate, where, the learning rate in the $k^{th}$ phase is denoted by $\eta_k$ and $P_k$ denotes the total number of data samples used in the $k^{th}$ phase. Similarly, for batch ramp schedules, $B_k$ denotes the batch size in the $k^{th}$ phase. For discussion, we will use the bias-variance decomposition terminology of risk (Jain et al., 2018; 2017; Zou et al., 2021; Wu et al., 2022a;b; Meterez et al., 2025). Informally, bias corresponds to the risk of the averaged iterates, while variance corresponds to the noise in the iterates, and $\mathcal{R}(\mathbf{w}_t) = \texttt{bias}_t + \texttt{variance}_t$. We will denote the stochastic gradient at time $t$ by $g_t$ and let $\mathbb{E}\|\mathbf{g}_t\|^2$ represent its expected squared norm under the population distribution.

## 5.1 MAIN RESULTS

In this section, we first introduce the main assumptions and discuss their implications, followed by the main theoretical results. Our first assumption states that the risk is almost non expansive, in the sense that at any point during training after starting the scheduling, the risk is close to the starting risk.

**Assumption 1** (Bounded risk.). *Suppose an SGD process and a given scheduling scheme, and let $t_0$ be the time where the scheduler starts. Then, we assume that there exists a constant $c > 1$ such that $\mathcal{R}(\mathbf{w}_t) \leq c\sigma^2$ for all $t > t_0$.*

In general, we expect every "well tuned" scheduler to start cutting when $\mathcal{R}(\mathbf{w}_{t_0}) \lesssim \sigma^2$, as we want to minimize the bias component of the risk before cutting down the learning rate to reduce noise in the iterates. Moreover, for a well-behaved schedule, as we expect the risk to decrease over time, this condition should hold throughout the process.

Our second assumption characterizes the gradient norms in the normalized SGD update rule.

**Assumption 2** (NSGD oracle access). *For normalized SGD, we assume access to an oracle that provides, at every step, the exact value of the expected squared gradient norms $\mathbb{E}\|\mathbf{g}_t\|^2$.*

In general, we don't have access to the ground truth gradient norms and rely on an exponential moving average - controlled by the $\beta_2$ hyperparameter in Adam, in order to estimate the gradient norms. Assumption 2 simplifies the analysis by giving us access to the true expected gradient squared norms. Our final assumption states that the expected gradient squared norms of the NSGD update rule are dominated by the additive noise term.

**Assumption 3** (Variance dominated.). *Assume that $\mathbb{E}\|\mathbf{g}_t\|^2 \asymp \frac{\sigma^2}{B_t}$.*

Under Assumption 3, the NSGD process effectively reduces to SGD with a rescaled learning rate, up to constant factors. Based on the previously established assumptions, we can now state the equivalence result. We use the notation $\mathcal{R}(\eta_t, B_t)$ to denote the risk at time $t$ of an SGD process trained with the learning rate scheduler $\eta$ and batch size scheduler $B_t$, where we omit the time subscript to denote constant learning rate or batch size respectively.

**Theorem 1** (SGD Equivalence). *Fix $\frac{0.01}{Tr(\mathbf{H})} \geq \eta > 0$, $B > 0$, and parameters $\alpha_1, \alpha_2 > 1$, $\beta_1, \beta_2 > 1$ with $\alpha_1\beta_1 = \alpha_2\beta_2$. Define the two phase-indexed schedules*

$$(\eta_k, B_k) := \left(\eta\, \alpha_1^{-k},\ B\, \beta_1^k\right), \qquad (\eta_k', B_k') := \left(\eta\, \alpha_2^{-k},\ B\, \beta_2^k\right), \quad k = 0, 1, 2, \dots$$

*and run two SGD procedures in phases $k = 0, 1, \dots$ so that, in phase $k$, each procedure processes the same number of samples (possibly depending on $k$) under its respective schedule. Let $\mathcal{R}(\eta_k, B_k)$ and $\mathcal{R}(\eta_k', B_k')$ denote the (population) risk of the two procedures at the end of phase $k$. If Assumption 1 holds (for both procedures) with constant $c$, then*

$$\mathcal{R}(1.01 \cdot \eta_k',\, B_k') \ \lesssim_c\ \mathcal{R}(\eta_k,\, B_k) \ \lesssim_c\ \mathcal{R}(\eta_k',\, B_k'),$$

*where $\mathcal{R}(\lambda \cdot \eta'_k, B'_k)$ denotes the risk of the second procedure when its entire learning-rate schedule is multiplied by a uniform factor $\lambda > 0$, and $A \lesssim_c B$ means $A \le C(c) B$ for a numerical constant $C(c)$ depending only on $c$ (and absolute constants).*

We defer the full proof to Appendix A.1. Now, we extend this result to Normalized SGD. Under Assumption 3, NSGD reduces to SGD with a rescaled learning rate $\tilde{\eta} \approx \eta \frac{\sqrt{B}}{\sigma \sqrt{\text{Tr}(\mathbf{H})}}$ (Equation equation 11). Consequently, we can extend Theorem 1 to the normalized SGD case. We formalize this in the following corollary:

**Corollary 1** (Normalized SGD Equivalence). *Fix $\frac{0.01}{Tr(\mathbf{H})} \ge \eta > 0$, $B > 0$, and parameters $\alpha_1, \alpha_2 > 1$, $\beta_1, \beta_2 > 1$ with $\alpha_1 \sqrt{\beta_1} = \alpha_2 \sqrt{\beta_2}$. Define the two phase-indexed schedules*

$$(\eta_k, B_k) := \left(\eta \, \alpha_1^{-k}, \, B \, \beta_1^k\right), \qquad (\eta'_k, B'_k) := \left(\eta \, \alpha_2^{-k}, \, B \, \beta_2^k\right), \quad k = 0, 1, 2, \dots$$

*and run two normalized SGD procedures in phases $k = 0, 1, \dots$ so that, in phase $k$, each procedure processes the same number of samples (possibly depending on $k$) under its respective schedule. Let $\mathcal{R}(\eta_k, B_k)$ and $\mathcal{R}(\eta'_k, B'_k)$ denote the (population) risk of the two procedures at the end of phase $k$. If Assumption 1 and 3 holds (for both procedures) with constant $c$, then*

$$\mathcal{R}(1.01 \cdot \eta'_k, \, B'_k) \ \lesssim_c \ \mathcal{R}(\eta_k, \, B_k) \ \lesssim_c \ \mathcal{R}(\eta'_k, \, B'_k),$$

*where $\mathcal{R}(\lambda \cdot \eta'_k, B'_k)$ denotes the risk of the second procedure when its entire learning-rate schedule is multiplied by a uniform factor $\lambda > 0$, and $A \lesssim_c B$ means $A \le C(c) B$ for a numerical constant $C(c)$ depending only on $c$ (and absolute constants).*

## 6 DISCUSSION AND CONCLUSIONS

In this work we have introduced Seesaw, a drop-in batch size and learning rate scheduler, theoretically motivated by optimization in quadratics using normalized SGD. We rigorously show that for stepwise schedulers there exists an equivalence between learning rate decay and batch size ramp-up, and empirically compare our scheduler with cosine annealing using a stepwise approximation of the cosine. Crucially, we also show that there exists a maximally aggressive batch size ramp up scheme without leading to instabilities and divergence during training. In the current implementation, Seesaw is able to decrease the serial runtime of a training run by $\approx 36\%$, bringing significant speedups to current pretraining pipelines. To conclude, we believe that our scheduler is a principled way of decreasing the runtime of any LLM pretraining run in an optimizer agnostic way.

## ACKNOWLEDGEMENTS

AM would like to thank Jacob Zavatone-Veth and Alex Damian for helpful discussions. The authors would also like to thank Max Shad and Bala Desinghu for their help with the cluster. AM, DM acknowledge the support of a Kempner Institute Graduate Research Fellowship. CP is supported by an NSF CAREER Award (IIS-2239780), DARPA grants DIAL-FP-038 and AIQ-HR00112520041, the Simons Collaboration on the Physics of Learning and Neural Computation, and the William F. Milton Fund from Harvard University. AM, SK and DM acknowledge that this work has been made possible in part by a gift from the Chan Zuckerberg Initiative Foundation to establish the Kempner Institute for the Study of Natural and Artificial Intelligence. SK and DM acknowledge support from the Office of Naval Research under award N0001422-1-2377 and the National Science Foundation Grant under award #IIS 2229881. DM is also supported by a Simons Investigator Fellowship, NSF grant DMS-2134157, DARPA grant W911NF2010021,and DOE grant DE-SC0022199.

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

# A  PROOFS FOR SECTION 5

## A.1  PRELIMINARIES

We take as a convention for eigenvalues ordering $\lambda_{\max} = \lambda_1 \geq \lambda_2 \geq \cdots > 0$. For two matrices $\mathbf{A}$ and $\mathbf{B}$ we use the notation $\mathbf{A} \preceq \mathbf{B}$ to denote that $\mathbf{B} - \mathbf{A}$ is positive semi-definite (PSD). We denote $\langle \mathbf{u}, \mathbf{v} \rangle$ for the inner product between $\mathbf{u}$ and $\mathbf{v}$. Moreover, with a slight abuse of notation, we use the notation $\leq$ as elementwise comparison, namely $\mathbf{u} \leq \mathbf{v}$ if $\mathbf{u}_i \leq \mathbf{v}_i$ for all $i$ and $\mathbf{A} \leq \mathbf{B}$ if $\mathbf{A}_{ij} \leq \mathbf{B}_{ij}$ for all $i, j$. To simplify the analysis, we will follow the approach of Meterez et al. (2025) and work in the eigenbasis of the data covariance $\mathbf{H}$. Denote the eigendecomposition of $\mathbf{H} = \mathbf{Q}\mathbf{\Lambda}\mathbf{Q}^\top$. For the sake of completeness, we restate the main derivation for the bias and variance iterates in the case of constant learning rate and constant batch size, starting from the SGD update rule:

$$\mathbf{w}_{t+1} - \mathbf{w}^\star = \left( \mathbf{I} - \frac{\eta}{B} \sum_{i=1}^{B} \mathbf{x}_i \mathbf{x}_i^\top \right)(\mathbf{w}_t - \mathbf{w}^\star) - \frac{\eta}{B} \sum_{i=1}^{B} \mathbf{x}_i \epsilon_i$$

$$\implies \mathbf{\Sigma}_{t+1} = \mathbf{\Sigma}_t - \eta\mathbf{\Sigma}_t\mathbf{H} - \eta\mathbf{H}\mathbf{\Sigma}_t + \eta^2 \left( 1 + \frac{1}{B} \right) \mathbf{H}\mathbf{\Sigma}_t\mathbf{H} + \frac{\eta^2}{B}\mathrm{Tr}(\mathbf{H}\mathbf{\Sigma}_t)\mathbf{H} + \frac{\eta^2}{B}\sigma^2\mathbf{I}$$

$$\implies \mathbf{M}_{t+1} = \mathbf{M}_t - \eta\mathbf{M}_t\mathbf{\Lambda} - \eta\mathbf{\Lambda}\mathbf{M}_t + \eta^2 \left( 1 + \frac{1}{B} \right) \mathbf{\Lambda}\mathbf{M}_t\mathbf{\Lambda} + \frac{\eta^2}{B}\mathrm{Tr}(\mathbf{\Lambda}\mathbf{M}_t)\mathbf{\Lambda} + \frac{\eta^2}{B}\sigma^2\mathbf{I} \quad (6)$$

where in the last equation $\mathbf{M}_t = \mathbf{Q}\mathbf{\Sigma}_t\mathbf{Q}^\top$ is the iterate covariance matrix rotated in the eigenbasis of $\mathbf{H}$. Since we can write the excess risk as:

$$\mathcal{R}(\mathbf{w}_t) - \mathcal{R}(\mathbf{w}^\star) = \frac{1}{2}\mathrm{Tr}(\mathbf{\Lambda}\mathbf{M}_t) = \frac{1}{2}\langle \lambda, \mathbf{m}_t \rangle$$

where $\mathbf{m}_t = \mathrm{diag}(\mathbf{M}_t)$, it suffices to push a diag operator through equation equation 6. Finally, we get:

$$\mathbf{m}_{t+1} = \underbrace{\left[ \mathbf{I} - 2\eta\mathbf{\Lambda} + \eta^2 \left( 1 + \frac{1}{B} \right) \mathbf{\Lambda}^2 + \frac{\eta^2}{B}\lambda\lambda^\top \right]}_{\mathbf{A}} \mathbf{m}_t + \frac{\eta^2\sigma^2}{B}\lambda = \mathbf{A}^t\mathbf{m}_0 + \frac{\eta^2\sigma^2}{B}\sum_{i=0}^{t-1}\mathbf{A}^i\lambda$$

where $\widetilde{\mathbf{m}}_t := \mathbf{A}^t\mathbf{m}_0$ and $\overline{\mathbf{m}}_t := \frac{\eta^2\sigma^2}{B}\sum_{i=0}^{t-1}\mathbf{A}^i\lambda$ are the bias and variance iterates respectively.

Before we begin proving the main statements, we introduce several helpful lemmas that we will use.

**Lemma 2.** *For $\eta \leq 0.01/Tr(\mathbf{H})$ and $\alpha \geq 1$, we have the elementwise inequality:*

$$\frac{\alpha^k}{\eta}\mathbf{1} \geq \left( \mathbf{I} - \left( \mathbf{I} - \frac{\eta}{\alpha^k}\mathbf{\Lambda} \right)^2 \right)^{-1} \lambda \geq \frac{\alpha^k}{2\eta}\mathbf{1}$$

*Proof.* We have:

$$\left( \mathbf{I} - \left( \mathbf{I} - \frac{\eta}{\alpha^k}\mathbf{\Lambda} \right)^2 \right)^{-1} = \left( \mathbf{I} - \left( \mathbf{I} + \frac{\eta^2}{\alpha^{2k}}\mathbf{\Lambda}^2 - 2\frac{\eta}{\alpha^2}\mathbf{\Lambda} \right) \right)^{-1}$$

$$= \left( \frac{\eta}{\alpha^k}\mathbf{\Lambda} \left( 2 - \frac{\eta}{\alpha^k}\mathbf{\Lambda} \right) \right)^{-1}$$

$$\geq \left( \frac{2\eta}{\alpha^k}\mathbf{\Lambda} \right)^{-1}$$

Note that trivially we also have the other direction by noticing that $\frac{1}{2 - \frac{\eta}{\alpha^k}\lambda} \leq 1$. Multiplying by $\lambda$ gives us the conclusion. □

**Lemma 3.** *For $\eta \leq 0.01/Tr(\mathbf{H})$ and $\alpha_1, \alpha_2, \beta_1, \beta_2 \geq 1$ such that $\alpha_1\beta_1 = \alpha_2\beta_2$ and $\alpha_1 \leq \alpha_2$, we have:*

$$\left( \mathbf{I} - \frac{1.01\,\eta}{\alpha_2^k}\mathbf{\Lambda} \right)^{2\beta_1^k} \preceq \left( \mathbf{I} - \frac{\eta}{\alpha_1^k}\mathbf{\Lambda} \right)^{2\beta_2^k} \preceq \left( \mathbf{I} - \frac{\eta}{\alpha_2^k}\mathbf{\Lambda} \right)^{2\beta_1^k}.$$

*Proof.* **RHS bound.** Since both sides are diagonal matrices, it suffices to prove the scalar inequality for every $x = \eta\lambda_i$:

$$\left(1 - \frac{x}{\alpha_1^k}\right)^{2\beta_2^k} \leq \left(1 - \frac{x}{\alpha_2^k}\right)^{2\beta_1^k}.$$

Taking logarithms and defining

$$f(x) = \frac{2\beta_2^k \log(1 - x/\alpha_1^k)}{2\beta_1^k \log(1 - x/\alpha_2^k)} = \frac{\alpha_1^k \log(1 - x/\alpha_1^k)}{\alpha_2^k \log(1 - x/\alpha_2^k)} = \frac{g(\alpha_1)}{g(\alpha_2)},$$

where $g(y) = y\log(1 - x/y)$. For $0 < x < 1$ and $y > 1$, $g(y)$ is monotonically increasing, so for $\alpha_1 \leq \alpha_2$, we have $g(\alpha_1) \leq g(\alpha_2)$ and hence $g(\alpha_1)/g(\alpha_2) \geq 1$ (since $g(\alpha_2) < 0$). Thus $f(x) \geq 1$, which proves the RHS inequality.

**LHS bound.** Similarly, we use the scalar inequality and the bounds

$$-x - \frac{x^2}{2} \geq \ln(1 - x) \geq -x - x^2.$$

Since $\ln(\cdot)$ is monotone, we apply it to both sides:

$$\beta_1^k \ln\left(1 - \frac{1.01}{\alpha_2^k}x\right) \leq \beta_1^k\left(-\frac{1.01}{\alpha_2^k}x - \frac{1.01^2}{2\alpha_2^{2k}}x^2\right),$$

$$\beta_2^k \ln\left(1 - \frac{1}{\alpha_1^k}x\right) \geq \beta_2^k\left(-\frac{1}{\alpha_1^k}x - \frac{1}{\alpha_1^{2k}}x^2\right).$$

It suffices to prove that:

$$\beta_1^k\left(-\frac{1.01}{\alpha_2^k}x - \frac{1.01^2}{2\alpha_2^{2k}}x^2\right) \geq \beta_2^k\left(-\frac{1}{\alpha_1^k}x - \frac{1}{\alpha_1^{2k}}x^2\right).$$

Using $\frac{\beta_1}{\alpha_2} = \frac{\beta_2}{\alpha_1}$ and $\frac{\beta_1}{\alpha_2^2} = \frac{\beta_2}{\alpha_1\alpha_2}$, we obtain:

$$\frac{1}{\alpha_1^k}(1.01) + \frac{1}{2\alpha_1^k\alpha_2^k}(1.01)^2 x - \frac{1}{\alpha_1^k} - \frac{1}{\alpha_1^{2k}}x \geq 0,$$

$$\iff x \leq \frac{0.01}{\frac{1}{\alpha_1^k} - \frac{1.01^2}{2\alpha_2^k}}.$$

Using $\alpha_1 \leq \alpha_2$, we get

$$x \leq \frac{\alpha_1^k \cdot 0.01}{1 - \frac{1.01^2}{2}},$$

which holds automatically under $\eta \leq 0.01/\text{Tr}(\mathbf{H})$ and $x = \eta\lambda_i$. This concludes the proof. $\square$

## A.2 Proofs of Main Statements

*Proof of Theorem 1.* Consider 2 processes: process 1 will have a learning rate step decay factor of $\alpha_1$ and a batch size ramp up factor of $\beta_1$ and process 2 will have $\alpha_2$ and $\beta_2$ respectively. Define the transition matrices:

$$\mathbf{A}_k = \left[\left(\mathbf{I} - \frac{\eta}{\alpha_1^k}\mathbf{\Lambda}\right)^2 + \frac{\eta^2}{B\alpha_1^{2k}\beta_1^k}(\mathbf{\Lambda}^2 + \lambda\lambda^\top)\right]$$

$$\mathbf{C}_k = \left[\left(\mathbf{I} - \frac{\eta}{\alpha_2^k}\mathbf{\Lambda}\right)^2 + \frac{\eta^2}{B\alpha_2^{2k}\beta_2^k}(\mathbf{\Lambda}^2 + \lambda\lambda^\top)\right]$$

Denote process 1 as $\mathbf{m}_k(\eta)$ and process 2 as $\mathbf{r}_k(\eta)$ where they depend on the base learning rate $\eta$ - note that we skip the indexing on $\eta$ when it is clear from context. In order to keep both the per stage data count, $\mathbf{m}_k$ does $\beta_2^k P_k$ steps per stage, and $\mathbf{r}_k$ does $\beta_1^k P_k$ steps per stage. We begin by establishing the upper bound first. Note that we assume that $\alpha_1\beta_1 = \alpha_2\beta_2$, and without loss of generality due to symmetry, that $\beta_1 \geq \beta_2$ (and consequently $\alpha_1 \leq \alpha_2$).

**Upper bound.** Before we begin, we introduce the idea behind the proof. We define $M_k = \beta_1^k P_k$ and $N_k = \beta_2^k P_k$. The derivation proceeds by unrolling the recurrence first over a single step, then over $\beta_2^k$ steps, and finally over $P_k$ stages.

$$
\mathbf{m}_{N_{1:k}} \leq \mathbf{A}_k \mathbf{m}_{N_{1:k}-1} + \frac{\eta^2 \sigma^2}{B \alpha_1^{2k} \beta_1^k} \lambda
$$

$$
\leq \left( \mathbf{I} - \frac{\eta}{\alpha_1^k} \boldsymbol{\Lambda} \right)^2 \mathbf{m}_{N_{1:k}-1} + (1 + 2c) \frac{\eta^2 \sigma^2}{B \alpha_1^{2k} \beta_1^k} \lambda,
$$

which follows from Assumption 1.

$$
\mathbf{m}_{N_{1:k}} \leq \left( \mathbf{I} - \frac{\eta}{\alpha_1^k} \boldsymbol{\Lambda} \right)^{2\beta_2^k} \mathbf{m}_{N_{1:k}-\beta_2^k} + (1 + 2c) \frac{\eta^2 \sigma^2}{B \alpha_1^{2k} \beta_1^k} \sum_{i=0}^{\beta_2^k - 1} \left( \mathbf{I} - \frac{\eta}{\alpha_1^k} \boldsymbol{\Lambda} \right)^{2i} \lambda
$$

$$
\leq \left( \mathbf{I} - \frac{\eta}{\alpha_1^k} \boldsymbol{\Lambda} \right)^{2\beta_2^k} \mathbf{m}_{N_{1:k}-\beta_2^k} + (1 + 2c) \frac{\eta^2 \sigma^2}{B \alpha_1^{2k} \beta_1^k} \left[ \mathbf{I} - \left( \mathbf{I} - \frac{\eta}{\alpha_1^k} \boldsymbol{\Lambda} \right)^{2\beta_2^k} \right] \left[ \mathbf{I} - \left( \mathbf{I} - \frac{\eta}{\alpha_1^k} \boldsymbol{\Lambda} \right)^2 \right]^{-1} \lambda.
$$

Applying Lemma 2, we have:

$$
\mathbf{m}_{N_{1:k}} \leq \left( \mathbf{I} - \frac{\eta}{\alpha_1^k} \boldsymbol{\Lambda} \right)^{2\beta_2^k} \mathbf{m}_{N_{1:k}-\beta_2^k} + (1 + 2c) \frac{\eta \sigma^2}{B \alpha_1^k \beta_1^k} \left[ \mathbf{I} - \left( \mathbf{I} - \frac{\eta}{\alpha_1^k} \boldsymbol{\Lambda} \right)^{2\beta_2^k} \right] \mathbf{1}
$$

$$
\leq \left( \mathbf{I} - \frac{\eta}{\alpha_1^k} \boldsymbol{\Lambda} \right)^{2\beta_2^k} \mathbf{m}_{N_{1:k}-\beta_2^k} + 2(1 + 2c) \frac{\eta^2 \sigma^2}{B} \left( \frac{\beta_2}{\alpha_1^2 \beta_1} \right)^k \lambda.
$$

By Lemma 3, we can replace the term with one involving $(\alpha_2, \beta_1)$:

$$
\mathbf{m}_{N_{1:k}} \leq \left( \mathbf{I} - \frac{\eta}{\alpha_2^k} \boldsymbol{\Lambda} \right)^{2\beta_1^k} \mathbf{m}_{N_{1:k}-\beta_2^k} + 2(1 + 2c) \frac{\eta^2 \sigma^2}{B} \left( \frac{\beta_2}{\alpha_1^2 \beta_1} \right)^k \lambda.
$$

Following, we can unroll over $P_k$:

$$
\mathbf{m}_{N_{1:k}} \leq \left( \mathbf{I} - \frac{\eta}{\alpha_2^k} \boldsymbol{\Lambda} \right)^{2M_k} \mathbf{m}_{N_{1:k-1}} + 2(1 + 2c) \frac{\eta^2 \sigma^2}{B} \left( \frac{\beta_2}{\alpha_1^2 \beta_1} \right)^k \sum_{i=0}^{P_k - 1} \left( \mathbf{I} - \frac{\eta}{\alpha_2^k} \boldsymbol{\Lambda} \right)^{2\beta_1^k i} \lambda.
$$

Finally, recursively unrolling across $k$ yields:

$$
\mathbf{m}_{N_{1:k}} \leq \left[ \prod_{s=1}^{k} \left( \mathbf{I} - \frac{\eta}{\alpha_2^s} \boldsymbol{\Lambda} \right)^{2M_s} \right] \mathbf{m}_0
$$

$$
+ 2(1 + 2c) \frac{\eta^2 \sigma^2}{B} \sum_{r=1}^{k} \left( \frac{1}{\alpha_1 \alpha_2} \right)^r \left[ \prod_{s=r+1}^{k} \left( \mathbf{I} - \frac{\eta}{\alpha_2^s} \boldsymbol{\Lambda} \right)^{2M_s} \right] \sum_{i=0}^{P_r - 1} \left( \mathbf{I} - \frac{\eta}{\alpha_2^k} \boldsymbol{\Lambda} \right)^{2\beta_1^r i} \lambda.
$$

For the lower bound, we follow a similar strategy, by bounding the term $\lambda\lambda^\top \geq 0$:

$$
\begin{aligned}
\mathbf{r}_{M_{1:k}} &\geq \left(\mathbf{I} - \frac{\eta}{\alpha_2^k}\mathbf{\Lambda}\right)^2 \mathbf{r}_{M_{1:k}-1} + \frac{\eta^2\sigma^2}{B\alpha_2^{2k}\beta_2^k}\lambda \\
&\geq \left(\mathbf{I} - \frac{\eta}{\alpha_2^k}\mathbf{\Lambda}\right)^{2\cdot\beta_1^k} \mathbf{r}_{M_{1:k}-\beta_1^k} + \frac{\eta^2\sigma^2}{B\alpha_2^{2k}\beta_2^k}\sum_{i=0}^{\beta_1^k-1}\left(\mathbf{I} - \frac{\eta}{\alpha_2^k}\mathbf{\Lambda}\right)^{2i}\lambda \\
&= \left(\mathbf{I} - \frac{\eta}{\alpha_2^k}\mathbf{\Lambda}\right)^{2\cdot\beta_1^k} \mathbf{r}_{M_{1:k}-\beta_1^k} + \frac{\eta^2\sigma^2}{B\alpha_2^{2k}\beta_2^k}\left[\mathbf{I} - \left(\mathbf{I} - \frac{\eta}{\alpha_2^k}\mathbf{\Lambda}\right)^{2\cdot\beta_1^k}\right]\left[\mathbf{I} - \left(\mathbf{I} - \frac{\eta}{\alpha_2^k}\mathbf{\Lambda}\right)^2\right]^{-1}\lambda \\
&\geq \left(\mathbf{I} - \frac{\eta}{\alpha_2^k}\mathbf{\Lambda}\right)^{2\cdot\beta_1^k} \mathbf{r}_{M_{1:k}-\beta_1^k} + \frac{1}{2}\frac{\eta\sigma^2}{B\alpha_2^k\beta_2^k}\left[\mathbf{I} - \left(\mathbf{I} - \frac{\eta}{\alpha_2^k}\mathbf{\Lambda}\right)^{2\cdot\beta_1^k}\right]\mathbf{1} \qquad \text{Lemma 2}\\
&\geq \left(\mathbf{I} - \frac{\eta}{\alpha_2^k}\mathbf{\Lambda}\right)^{2\cdot\beta_1^k} \mathbf{r}_{M_{1:k}-\beta_1^k} + \frac{1}{4}\frac{\eta^2\sigma^2}{B}\left(\frac{\beta_1}{\alpha_2^2\beta_2}\right)^k\lambda \\
&\geq \left(\mathbf{I} - \frac{\eta}{\alpha_2^k}\mathbf{\Lambda}\right)^{2\cdot M_k} \mathbf{r}_{M_{1:k-1}} + \frac{1}{4}\frac{\eta^2\sigma^2}{B}\left(\frac{\beta_1}{\alpha_2^2\beta_2}\right)^k\sum_{i=0}^{P_k-1}\left(\mathbf{I} - \frac{\eta}{\alpha_2^k}\mathbf{\Lambda}\right)^{2\beta_1^k i}\lambda \\
&\geq \left[\prod_{s=1}^k\left(\mathbf{I} - \frac{\eta}{\alpha_2^s}\mathbf{\Lambda}\right)^{2\cdot M_s}\right]\mathbf{r}_0 \\
&\quad + \frac{1}{4}\frac{\eta^2\sigma^2}{B}\sum_{r=1}^k\left(\frac{1}{\alpha_1\alpha_2}\right)^r\left[\prod_{s=r+1}^k\left(\mathbf{I} - \frac{\eta}{\alpha_2^s}\mathbf{\Lambda}\right)^{2\cdot M_s}\right]\sum_{i=0}^{P_r-1}\left(\mathbf{I} - \frac{\eta}{\alpha_2^k}\mathbf{\Lambda}\right)^{2\beta_1^r i}\lambda
\end{aligned}
$$

Note that the bias terms are equal $\widetilde{\mathbf{r}}_{M_{1:k}} = \widetilde{\mathbf{m}}_{N_{1:k}}$, and the variance terms are $\overline{\mathbf{m}}_{N_{1:k}} \geq 4(1 + 2c)\overline{\mathbf{r}}_{M_{1:k}}$. Dotting the terms into $\lambda$ gives us the upper bound from Theorem 1.

**Lower bound.** We now turn our attention towards proving the lower bound in Theorem 1. Note that the bias terms have an exponentially decaying dominating term. In order to obtain an inequality in the reverse direction for these terms, we compare $\mathbf{m}(\eta)$ with $\mathbf{r}(1.01\eta)$. We begin with lower bounding $\mathbf{m}$:

$$
\begin{aligned}
\mathbf{m}_{N_{1:k}}(\eta) &\geq \left(\mathbf{I} - \frac{\eta}{\alpha_1^k}\mathbf{\Lambda}\right)^2\mathbf{m}_{N_{1:k}-1} + \frac{\eta^2\sigma^2}{B\alpha_1^{2k}\beta_1^k}\lambda \\
&\geq \left(\mathbf{I} - \frac{\eta}{\alpha_1^k}\mathbf{\Lambda}\right)^{2\beta_2^k}\mathbf{m}_{N_{1:k}-\beta_2^k} + \frac{1}{4}\frac{\eta^2\sigma^2}{B}\left(\frac{1}{\alpha_1\alpha_2}\right)^k\lambda \\
&\geq \left(\mathbf{I} - \frac{\eta}{\alpha_1^k}\mathbf{\Lambda}\right)^{2N_k}\mathbf{m}_{N_{1:k-1}} + \frac{1}{4}\frac{\eta^2\sigma^2}{B}\left(\frac{1}{\alpha_1\alpha_2}\right)^k\sum_{i=0}^{P_k-1}\left(\mathbf{I} - \frac{\eta}{\alpha_1^k}\mathbf{\Lambda}\right)^{2\beta_2^k i}\lambda \\
&\geq \left[\prod_{s=1}^k\left(\mathbf{I} - \frac{\eta}{\alpha_1^s}\mathbf{\Lambda}\right)^{2N_s}\right]\mathbf{m}_0 \\
&\quad + \frac{1}{4}\frac{\eta^2\sigma^2}{B}\sum_{r=1}^k\left(\frac{1}{\alpha_1\alpha_2}\right)^r\left[\prod_{s=r+1}^k\left(\mathbf{I} - \frac{\eta}{\alpha_1^s}\mathbf{\Lambda}\right)^{2N_s}\right]\sum_{i=0}^{P_r-1}\left(\mathbf{I} - \frac{\eta}{\alpha_1^k}\mathbf{\Lambda}\right)^{2\beta_2^r i}\lambda
\end{aligned}
$$

Now we need to establish an upper bound for $\mathbf{r}(1.01\eta)$. We follow a similar analysis as we did for the upper bound subsection:

$$\mathbf{r}_{M_{1:k}}(1.01\eta)$$

$$\leq \left(\mathbf{I} - \frac{1.01\eta}{\alpha_2^k}\mathbf{\Lambda}\right)^2 \mathbf{r}_{M_{1:k}-1} + 1.01^2 \cdot (1+2c)\frac{\eta^2\sigma^2}{B\alpha_1^{2k}\beta_1^k}\lambda$$

$$\leq \left(\mathbf{I} - \frac{1.01\eta}{\alpha_2^k}\mathbf{\Lambda}\right)^{2\beta_1^k} \mathbf{r}_{M_{1:k}-\beta_1^k} + 2 \cdot 1.01^2 \cdot (1+2c)\frac{\eta^2\sigma^2}{B}\left(\frac{1}{\alpha_1\alpha_2}\right)^k \lambda$$

$$\leq \left(\mathbf{I} - \frac{\eta}{\alpha_1^k}\mathbf{\Lambda}\right)^{2\beta_2^k} \mathbf{r}_{M_{1:k}-\beta_1^k} + 2 \cdot 1.01^2 \cdot (1+2c)\frac{\eta^2\sigma^2}{B}\left(\frac{1}{\alpha_1\alpha_2}\right)^k \lambda \qquad \text{Lemma 3}$$

$$\leq \left(\mathbf{I} - \frac{\eta}{\alpha_1^k}\mathbf{\Lambda}\right)^{2N_k} \mathbf{r}_{M_{1:k-1}} + 2 \cdot 1.01^2 \cdot (1+2c)\frac{\eta^2\sigma^2}{B}\left(\frac{1}{\alpha_1\alpha_2}\right)^k \sum_{i=0}^{P_k-1}\left(\mathbf{I} - \frac{\eta}{\alpha_1^k}\mathbf{\Lambda}\right)^{2\beta_2^k i}\lambda$$

$$\leq \left[\prod_{s=1}^{k}\left(\mathbf{I} - \frac{\eta}{\alpha_1^s}\mathbf{\Lambda}\right)^{2N_s}\right]\mathbf{r}_0$$

$$+ 2 \cdot 1.01^2 \cdot (1+2c)\frac{\eta^2\sigma^2}{B}\sum_{r=1}^{k}\left(\frac{1}{\alpha_1\alpha_2}\right)^r\left[\prod_{s=r+1}^{k}\left(\mathbf{I} - \frac{\eta}{\alpha_1^s}\mathbf{\Lambda}\right)^{2N_s}\right]\sum_{i=0}^{P_r-1}\left(\mathbf{I} - \frac{\eta}{\alpha_1^k}\mathbf{\Lambda}\right)^{2\beta_2^r i}\lambda$$

Comparing the bias and variance terms gives us the conclusion. $\qquad\square$

# B  NORMALIZED SGD ANALYSIS

Under the setup introduced in Section 5, we have the update rule for normalized SGD is:

$$\mathbf{w}_{t+1} = \mathbf{w}_t - \eta \frac{1}{\sqrt{\mathbb{E}\|\mathbf{g}_t\|^2}} \mathbf{g}_t$$

where $\mathbf{g}_t = \frac{1}{B} \sum_{i=1}^{B} \mathbf{g}_t^{(i)}$ for $i$ indexing the sample and batch size $B$.

For MSE and $y = (\mathbf{w}^\star)^\top \mathbf{x} + \epsilon$, the loss is:

$$\mathcal{L}(\mathbf{w}_t) = \frac{1}{2B} \sum_{i=1}^{B} (\mathbf{w}_t^\top \mathbf{x}^{(i)} - y^{(i)})^2$$

$$= \frac{1}{2B} \sum_{i=1}^{B} ((\mathbf{w}_t - \mathbf{w}^\star)^\top \mathbf{x}^{(i)} - \epsilon)^2$$

If we look at the risk at time $t$ we have:

$$\mathcal{R}(\mathbf{w}_t) = \frac{1}{2B} \sum_{i=1}^{B} \mathbb{E}[(\mathbf{w}_t - \mathbf{w}^\star)^\top \mathbf{x}^{(i)} \mathbf{x}^{(i),\top} (\mathbf{w}_t - \mathbf{w}^\star) + \epsilon^2] \tag{7}$$

$$= \frac{1}{2B} \sum_{i=1}^{B} \mathbb{E}[(\mathbf{w}_t - \mathbf{w}^\star)^\top \mathbf{x}^{(i)} \mathbf{x}^{(i),\top} (\mathbf{w}_t - \mathbf{w}^\star)] + \frac{\sigma^2}{2} \tag{8}$$

$$= \frac{1}{2} \mathbb{E}[(\mathbf{w}_t - \mathbf{w}^\star)^\top \mathbf{x}\mathbf{x}^\top (\mathbf{w}_t - \mathbf{w}^\star)] + \frac{\sigma^2}{2} \tag{9}$$

$$= \frac{1}{2} \text{Tr}(\mathbf{H}\boldsymbol{\Sigma}_t) + \frac{\sigma^2}{2} \tag{10}$$

So the risk is equal to:

$$\mathcal{R}(\mathbf{w}_t) = \frac{1}{2} \text{Tr}(\mathbf{H}\boldsymbol{\Sigma}_t) + \frac{\sigma^2}{2} \implies \mathcal{R}(\mathbf{w}_t) - \mathcal{R}(\mathbf{w}^\star) = \frac{1}{2} \text{Tr}(\mathbf{H}\boldsymbol{\Sigma}_t)$$

**Analyzing the gradients**  Taking the gradient for 1 sample:

$$\mathbf{g}_t^{(i)} := \nabla_{\mathbf{w}_t} \mathcal{L}^{(i)} = (\mathbf{w}_t^\top \mathbf{x}^{(i)} - y^{(i)})\mathbf{x}^{(i)} = \mathbf{x}^{(i)}(\mathbf{x}^{(i)})^\top (\mathbf{w}_t - \mathbf{w}^\star) - \epsilon^{(i)}\mathbf{x}^{(i)}.$$

So we have:

$$\mathbf{g}_t = \frac{1}{B} \sum_{i=1}^{B} \mathbf{x}^{(i)}(\mathbf{x}^{(i)})^\top (\mathbf{w}_t - \mathbf{w}^\star) - \frac{1}{B} \sum_{i=1}^{B} \epsilon^{(i)}\mathbf{x}^{(i)}.$$

Moving forwards, we need to calculate the term in the denominator. Skipping the time index when convenient in order to simplify the notation, and writing $\delta_t := \mathbf{w}_t - \mathbf{w}^\star$, we compute $\mathbb{E}\|\mathbf{g}\|^2$ by conditioning on $\mathbf{w}_t$ and then applying the tower rule:

$$\mathbb{E}\|\mathbf{g}\|^2 = \mathbb{E}\Big[\mathbb{E}\big[\|\mathbf{g}\|^2 \mid \mathbf{w}_t\big]\Big],$$

$$\mathbb{E}\big[\|\mathbf{g}\|^2 \mid \mathbf{w}_t\big] = \frac{1}{B^2} \sum_{i=1}^{B} \sum_{j=1}^{B} \mathbb{E}\big[\delta_t^\top \mathbf{x}^{(i)}(\mathbf{x}^{(i)})^\top \mathbf{x}^{(j)}(\mathbf{x}^{(j)})^\top \delta_t \mid \mathbf{w}_t\big]$$

$$+ \frac{1}{B^2} \sum_{i=1}^{B} \sum_{j=1}^{B} \mathbb{E}\big[\epsilon^{(i)}\epsilon^{(j)}(\mathbf{x}^{(i)})^\top \mathbf{x}^{(j)} \mid \mathbf{w}_t\big].$$

**First term** Let $\mathbf{H} := \mathbb{E}[\mathbf{x}\mathbf{x}^\top]$. For $i \neq j$, independence gives

$$\mathbb{E}\Big[\mathbf{x}^{(i)}(\mathbf{x}^{(i)})^\top \mathbf{x}^{(j)}(\mathbf{x}^{(j)})^\top\Big] = \mathbf{H}^2.$$

For $i = j$, assume $\mathbf{x} \sim \mathcal{N}(0, \mathbf{H})$, so that the Gaussian fourth-moment identity yields

$$\mathbb{E}[\mathbf{x}\mathbf{x}^\top \mathbf{x}\mathbf{x}^\top] = \mathrm{Tr}(\mathbf{H})\mathbf{H} + 2\mathbf{H}^2.$$

Therefore,

$$\mathbb{E}\big[\delta_t^\top \mathbf{x}^{(i)}(\mathbf{x}^{(i)})^\top \mathbf{x}^{(j)}(\mathbf{x}^{(j)})^\top \delta_t \mid \mathbf{w}_t\big] = \delta_t^\top \Big(\mathbf{H}^2(1 - \delta_{ij}) + \delta_{ij}\big(\mathrm{Tr}(\mathbf{H})\mathbf{H} + 2\mathbf{H}^2\big)\Big)\delta_t.$$

Summing over $i, j$ yields

$$\frac{1}{B^2}\sum_{i=1}^{B}\sum_{j=1}^{B}\mathbb{E}\big[\delta_t^\top \mathbf{x}^{(i)}(\mathbf{x}^{(i)})^\top \mathbf{x}^{(j)}(\mathbf{x}^{(j)})^\top \delta_t \mid \mathbf{w}_t\big] = \delta_t^\top \Big(\frac{1}{B}\big(\mathrm{Tr}(\mathbf{H})\mathbf{H} + 2\mathbf{H}^2\big) + \Big(1 - \frac{1}{B}\Big)\mathbf{H}^2\Big)\delta_t$$

$$= \delta_t^\top \Big(\frac{1}{B}\mathrm{Tr}(\mathbf{H})\mathbf{H} + \Big(1 + \frac{1}{B}\Big)\mathbf{H}^2\Big)\delta_t.$$

**Second term** Using $\mathbb{E}[\epsilon^{(i)}\epsilon^{(j)}] = \sigma^2\delta_{ij}$ and $\mathbb{E}[(\mathbf{x}^{(i)})^\top \mathbf{x}^{(j)}] = \mathrm{Tr}(\mathbf{H})\delta_{ij}$, we get

$$\frac{1}{B^2}\sum_{i=1}^{B}\sum_{j=1}^{B}\mathbb{E}\big[\epsilon^{(i)}\epsilon^{(j)}(\mathbf{x}^{(i)})^\top \mathbf{x}^{(j)} \mid \mathbf{w}_t\big] = \frac{1}{B^2}\sum_{i=1}^{B}\sigma^2\mathbb{E}[(\mathbf{x}^{(i)})^\top \mathbf{x}^{(i)}] = \sigma^2\frac{\mathrm{Tr}(\mathbf{H})}{B}.$$

So altogether, conditional on $\mathbf{w}_t$,

$$\mathbb{E}\big[\|\mathbf{g}\|^2 \mid \mathbf{w}_t\big] = \delta_t^\top \Big(\frac{1}{B}\mathrm{Tr}(\mathbf{H})\mathbf{H} + \Big(1 + \frac{1}{B}\Big)\mathbf{H}^2\Big)\delta_t + \sigma^2\frac{\mathrm{Tr}(\mathbf{H})}{B}.$$

Taking the outside expectation and letting $\mathbf{\Sigma}_t := \mathbb{E}[\delta_t\delta_t^\top]$, we obtain

$$\mathbb{E}\|\mathbf{g}_t\|^2 = \mathrm{Tr}\Big(\Big(\frac{1}{B}\mathrm{Tr}(\mathbf{H})\mathbf{H} + \Big(1 + \frac{1}{B}\Big)\mathbf{H}^2\Big)\mathbf{\Sigma}_t\Big) + \sigma^2\frac{\mathrm{Tr}(\mathbf{H})}{B}$$

$$= \frac{1}{B}\mathrm{Tr}(\mathbf{H})\mathrm{Tr}(\mathbf{H}\mathbf{\Sigma}_t) + \Big(1 + \frac{1}{B}\Big)\mathrm{Tr}(\mathbf{H}^2\mathbf{\Sigma}_t) + \sigma^2\frac{\mathrm{Tr}(\mathbf{H})}{B}.$$

Since $\mathbf{\Sigma}_t \preceq \mathcal{O}(\sigma^2\mathbf{I})$ (Lemma 8) (Jain et al., 2018), then for large enough $t$, we have that the gradient norms are dominated by the additive variance, which is captured in Assumption 3. For the remainder of this paper we will assume $t$ is large enough for this assumption to hold, and with an abuse of notation we will write $=$ (as opposed to $\approx$):

$$\mathbb{E}\|\mathbf{g}_t\|^2 = \frac{\sigma^2}{B}\mathrm{Tr}(\mathbf{H}).$$

Under Assumption 3, we have the following update rule:

$$\mathbf{w}_{t+1} = \mathbf{w}_t - \eta\frac{\sqrt{B}}{\sigma\sqrt{\mathrm{Tr}(\mathbf{H})}}\nabla_{\mathbf{w}_t}\mathcal{L} \tag{11}$$

Note that this is simply SGD with a learning rate $\tilde{\eta} = \eta\frac{\sqrt{B}}{\sigma\sqrt{\mathrm{Tr}(\mathbf{H})}}$.

## B.1 HOW AGGRESIVE CAN THE SCHEDULER BE?

In this section we provide a short lemma explaining what is the most aggressive scheduler we could possibly used, based on hard contraints on $\alpha, \beta$.

**Lemma 4** (Divergence conditions.). *Suppose we are in the same setting as Corollary 1. For a fixed initial learning rate $\eta$, the training dynamics diverge asymptotically if $\alpha < \sqrt{\beta}$ as the training time goes to infinity, for $\alpha$ and $\beta$ constants independent of time.*

*Proof.* To see this, we focus on the scaling of $\tilde{\eta}_k \approx \eta\left(\frac{\sqrt{\beta}}{\alpha}\right)^k$. Note that if $\sqrt{\beta} > \alpha$, then at every cut we are effectively increasing the learning rate. Thus, there must exist $k > 0$ such that $\tilde{\eta}_k > \eta_{\max}$, where $\eta_{\max}$ is the maximum convergent learning rate for SGD (Wu et al., 2022b; Jain et al., 2018), leading to divergence. □

## C    COMPARISON TO OTHER SCHEDULERS

We compare our scheme with other schedulers in this section.

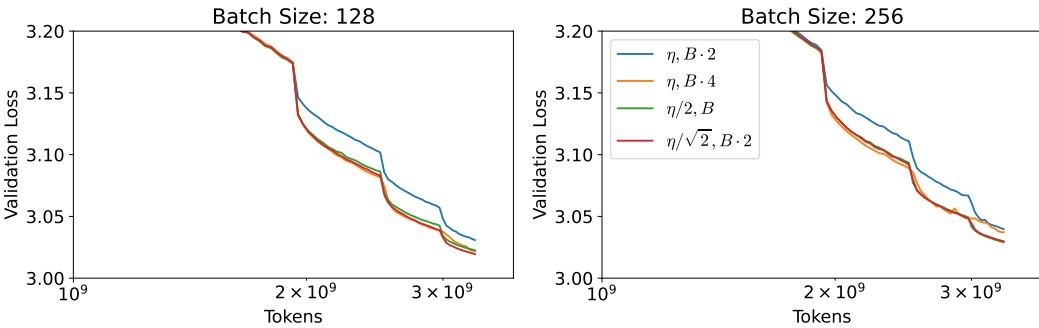

Figure 4: 150M models trained with 4 different schedules, at CBS (right) and just below (left). Blue trace keeps learning rate fixed and doubles batch size, orange trace keeps learning rate fixed and quadruples batch size, green trace halves learning rate at fixed batch size, and red trace is Seesaw. Note that the naive scheduling (blue) severely underperforms the baseline (green) and Seesaw (red).

# D  AUXILIARY LOSSES

In this section we ablate over the effect of z-loss on the training dynamics (OLMo et al., 2024). We observe no difference in the training stability of our models at 150M scale in Figure 5:

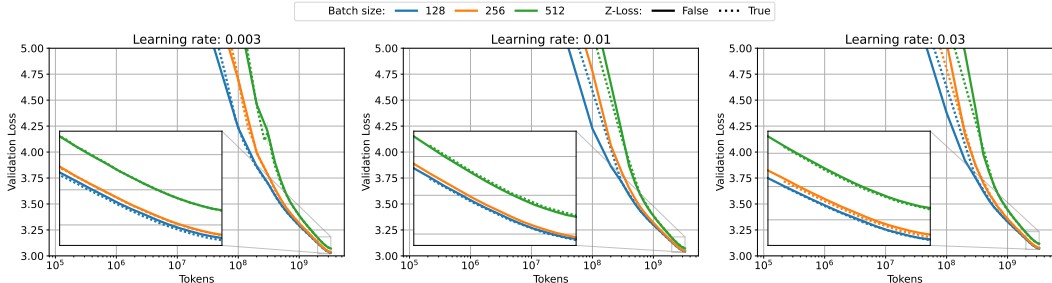

Figure 5: 150M models trained with cosine decay in Chinchilla scale, across 3 learning rates and 3 batch sizes. Note that the final validation losses are equal whether Z-Loss is enabled or not.

However, while the final validation loss does not change as an effect of z-loss at our scale, we have observed certain instabilities in the z-loss towards the end of training when using Seesaw in Figure 6. We speculate that the way we are scaling the learning rate and batch size might not be the proper way to do it for z-loss, and we leave this study for future work.

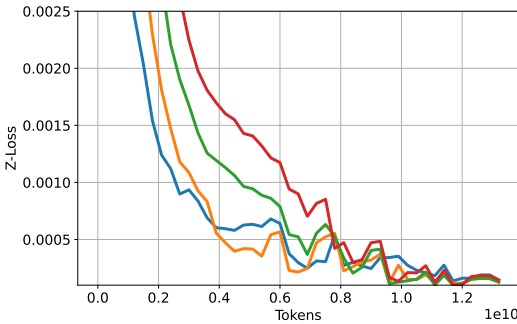

Figure 6: 600M models trained with Seesaw decay in Chinchilla scale, with Z-Loss.

# E    WEIGHT DECAY

In this section we provide experiments on 150M models trained with AdamW, sweeping weight decay $\lambda \in \{0.000001, 0.00001, 0.0001, 0.001, 0.01, 0.1, 1.0\}$ and learning rate $\eta \in \{0.001, 0.003, 0.01, 0.03\}$, and the rest of the parameters are as explained in Section 4. For every figure we pick the best $(\eta, \lambda)$ pair on cosine annealing, and we use the values for Seesaw. Across all batch sizes (128, 256, 512), the optimal $(\eta, \lambda)$ pair from the sweep turned out to be $(\eta, \lambda) = (0.003, 0.0001)$. Figure 7 shows the results:

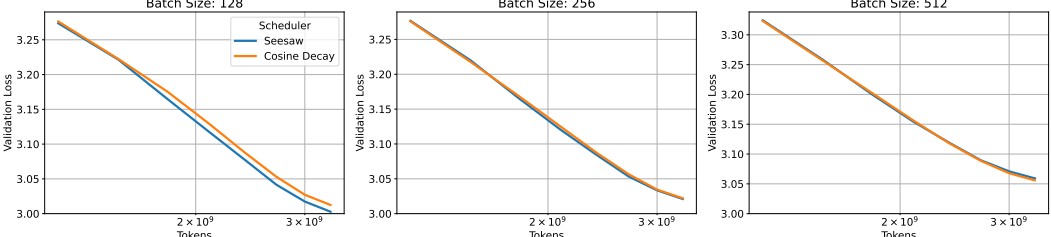

Figure 7: 150M experiments with weight decay across different batch sizes (128, 256, 512) for cosine annealing and Seesaw, for learning rate and weight decay values $(\eta, \lambda) = (0.003, 0.0001)$. Note that the losses overlap during training. We provide the final validation losses in Table 3.

Table 3 shows the final validation losses:

|  | B=128 | B=256 | B=512 |
|---|---|---|---|
| 150M (cosine) | 3.0125 | 3.0220 | 3.0559 |
| 150M (Seesaw) | 3.0027 | 3.0210 | 3.0588 |

Table 3: Final validation losses picked at the best learning rate (for the cosine annealing scheduler) for each batch size, for $\alpha = 1.1$ and weight decay 0.003. Note that the dynamics match robustly.

# F  OVERTRAINED RUNS

In this section we provide experiments for 150M models in the overtrained regime. We train for $4\times$ Chinchilla (so approximately 13.2B tokens), while sweeping over learning rates and batch sizes in the same range as Section 4. We show in Figure 8, and the final losses of these runs in Table 4, where the plots are done at the optimal learning rate for cosine.

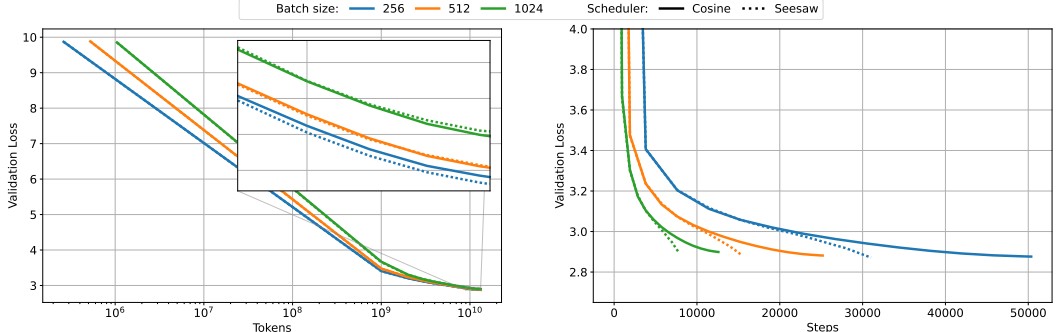

Figure 8: Seesaw comparison with cosine decay in 150M models trained at $4\times$ Chinchilla scale. For more experimental details, see Section 4. Note that the schedulers agree in the final losses, with the actual values shown in Table 4.

|  | B=256 | B=512 | B=1024 |
|---|---|---|---|
| 150M (cosine) | 2.8762 | 2.8814 | 2.8990 |
| 150M (Seesaw) | 2.8724 | 2.8820 | 2.9016 |

Table 4: Final validation losses for 150M models trained at $4\times$ Chinchilla.

# G ADDITIONAL FIGURES

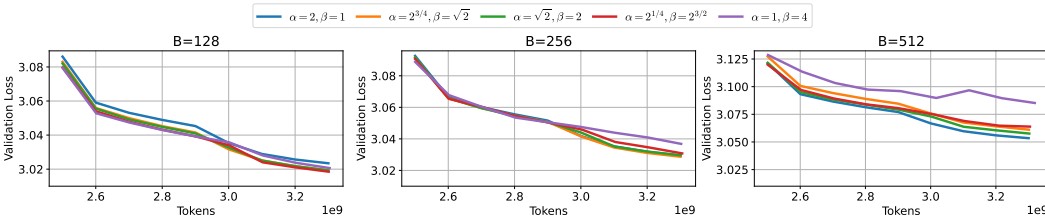

Figure 9: 150M models trained at batch size 128, 256, 512 with $\alpha$ and $\beta$ values following the line of equivalence $\alpha\sqrt{\beta} = 2$ described in Table 2.

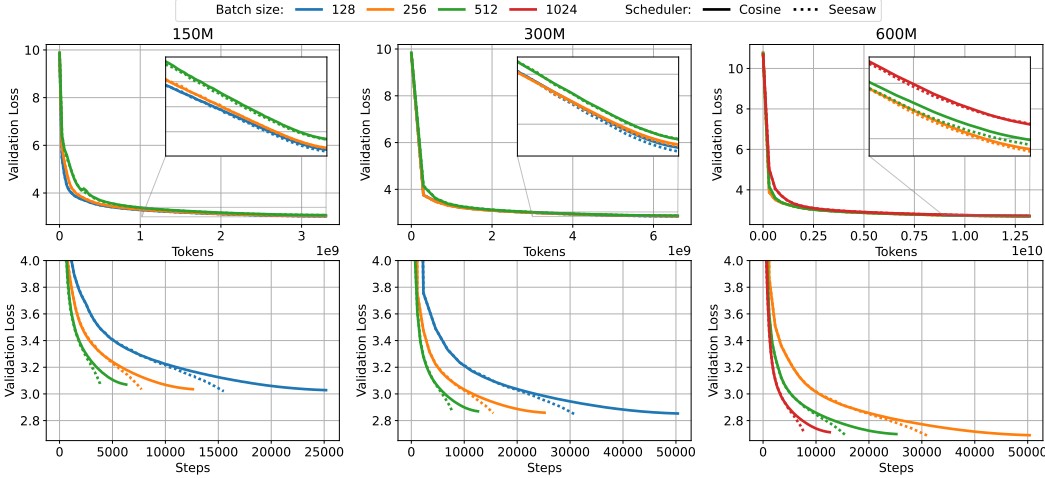

Figure 10: Seesaw comparison with cosine decay in 150M (left), 300M (middle) and 600M (right) models trained at Chinchilla scale. The validation losses at the end of training are provided in Table 1. For more experimental details, see Section 4.

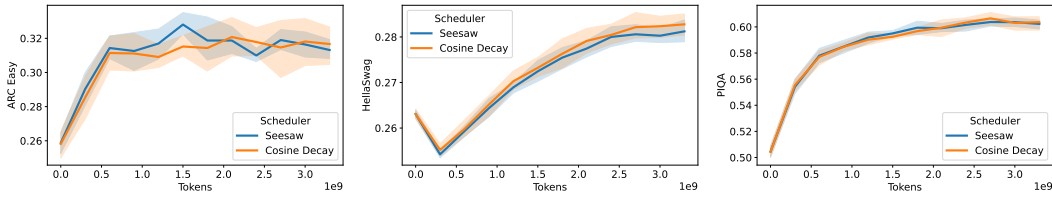

Figure 11: Downstream evals comparison with cosine decay in 150M models between Seesaw and Cosine decay trained at CBS (256) for $1\times$ Chinchilla, at the optimal learning rate for cosine. Note that the 2 methods have similar performance. The shades represent standard deviations over 5 seeds, taken due to the noisy nature of the evals.

