# OpenReview forum: "Seesaw: Accelerating Training by Balancing Batch Size and Learning Rate Scheduling"
_ICLR.cc/2026/Conference — ICLR 2026 Poster_

### Official Review · Reviewer_gioo · 2025-10-25

**Soundness:** 1
**Presentation:** 1
**Contribution:** 2
**Rating:** 2
**Confidence:** 5

**Summary:**

This paper introduces a technique to trade off a decrease in LR for an increase in batch size for LLM training.  The recipe is an application of the square-root scaling rule previously proposed for Adam training: whenever you double the batchsize, you should increase the LR by \sqrt(2).  Here, the idea is that when we would normally drop the LR by 2 according to its decay schedule, we immediately apply the square-root scaling rule, with the net effect that the LR gets decreased by only \sqrt(2), and the batch size is doubled.  Experiments on 150M, 300M, and 600M models trained to a compute-efficient 20 tokens-per-parameter show this approach can equal the loss of the baseline recipe, but using fewer steps (which you can calculate in advance).

**Strengths:**

It makes sense to revisit ideas like "Don't decay the LR, increase the batch size" in the context of modern LLMs.  Companies are competing to train the next generation of models as quickly as possible; if we can train faster, without spending additional FLOPs, it's of definite benefit.

Testing a simple recipe, as in the paper, is definitely the next step; if it works, people can immediately apply it.  Moreover, the ability to deterministically calculate the number of steps in advance is useful for resource planning.  I.e., the schedule is not based on, e.g., online measurements of the gradient noise scale as in https://arxiv.org/abs/2411.00999, but rather are motivated in advance from the LR schedule.

I would absolutely encourage the authors to keep working in this direction!

**Weaknesses:**

In its current form, the paper is only half-baked, in terms of writing, experimental rigor, theory, awareness of prior work, etc.  The paper gave me the feeling that the ICLR reviewers were doing the work of proofreading the paper, rather than mentors or co-authors, which was unsettling.

- The idea that Adam is basically normalized SGD, and that this necessitates dividing the learning rate by the square root of the batch size adjustment, was previously articulated in, e.g., in "Hilton - Batch size-invariance for policy optimization - 2110.00641v3", where (citing Hardin 2017), they similarly note "Adam divides the gradient by a running estimate of the root mean square gradient".  Hilton et al. divide this estimate into the square root of {gradient-mean-squared plus gradient-variance}, and show how the batch size reduces the variance, and essentially leads to the square root LR adjustment.  Since this paper wasn't discussed or cited, I'm not sure what the connections are, or the extent to which the theory as presented goes beyond this.

- Regarding soundness, I was concerned why we tested the extreme values of the equivalence (Figure 2) at 2x the CBS?  Especially since they're much closer at 256, it makes me suspect that they're even closer at 128, yet this wasn't presented.

- Also regarding soundness: I just feel there isn't the depth and breadth of experiments that we would normally find in an ICLR paper.  Beyond this, I feel like there's not enough depth and breadth to convince me to use this approach in my own training.

- Limitations were not discussed, e.g., not acknowledging that state-of-the-art models are trained with non-zero weight decay, to higher tokens-per-parameter, etc.  That this was only on one dataset with one tokenizer with one optimizer, etc.

- The claim in the abstract that we "*approach the theoretical limit implied by our analysis*," is misleading, as approaching this limit is not, like, a sign the model is training well or whatever, but just that the continuous-time approximation that you used returns a number that is close to the actual number of steps, right?  I mean, you could run a simple dry-run simulation to exactly determine how many steps the model will take given your algorithm, it's not like "abstract-level-ITALICIZED-claim" significant that your continuous-limit version provides a close answer, right?  The significant thing is that the losses match, but, like I said, that's got nothing to do your continuous-time version.  Unless I'm really misunderstanding something, which is possible, because...

The paper is poorly written and organized, and not from like a non-native-speaker grammar perspective, but from a thinking-and-planning-out-the-paper-clearly perspective.  Some points on these lines:

- It's confusing as heck to use α and β to represent the adjustments to the LR and batch size --- used in the product that should remain constant --- and to use them as actual values that get substituted into this equation, essentially α := √α, β := (√α)^2. You know what I mean?  Like, setting β = α is needed in order to satisfy α = √β.  It's almost non-sensical.

- "where the schedulers are equivalent in terms of loss as long as we keep the product α√β fixed" – what schedulers???  The one where B doesn’t change and the one where it does?
  - Let’s use α and β to be the adjustments to the LR and batch size, respectively.
  - Case 1: α = C and β = 1.  I.e., the points where the LR schedule would drop by C and we don’t change the batch size.  α√β = C * √1 = C.  (e.g., C=2 would give us the abstract of the paper)
  - Case 2 (proposed): α = √C, and β = something.  To maintain the invariant, β = C, only this way will α√β = √C√C = C as before.
  - As far as I can tell, that’s our only option for β in order to keep the product fixed.  But then we say, of all the ways we could adjust the batch size, i.e., of all the β values, the most aggressive we can use must satisfy α = √β, i.e., β = α^2.  So if α = √C, β = C.  But isn’t this our only option to have an equivalent scheduler?
  - I REALLY wanted to understand what you’re saying here, but I just could not.  Maybe on a second reading of the paper... but I'm just a reviewer, make my life easier please!

- Not enough context prior to the experiments
  - E.g., there's a section on “Assumption 3”, but this assumption is only mentioned parenthetically before this point

- I don’t really understand why all the content that was collected into Section 3 is in there:
  - Intuition
  - Theoretical results, with lots of pointers to other parts of the paper.
  - The formal algorithm
  - Like, you say, “the NSGD update rule, which is a crucial component of designing Seesaw”, but it seems to me that using a simple square-root scaling rule would have been sufficient here.

- When you present your main findings (Table 1), maybe provide some interpretation?  Like, what is the take-home message here?

- Too much required for understanding is in the appendices.

I could go on.

Nitpicks:
- For the Taylor expansions, it would have been helpful to define x_0, x_1 and x_2, and to define the noise terms precisely.  Should there be some expectations here?  Can you cite the specific section of Malladi that “shows this argument”?
- Might be worth pointing out somewhere that actually changing the batch size in a fine-grained way is challenging for modern large-scale LLM GPU deployments… although perhaps not on other hardware…
- “Understanding batch size ramp up schemes during training has been a topic of interest in recent years”.  Really?  I mean, people have used it, but has "*understanding* ramp-up" been a topic of interest?
- Wrong use of \citet vs \citep, e.g., “Recently, (Meterez et al., 2025) have used” should instead by \citet
- Typo: “We further empirically comapre* Seesaw”
- Table 1: “Note that the dynamics match” --- do you mean the final losses?  Might be good to highlight (e.g., color/italicize/shade/bolden) the CBS cells somehow.
- Says "trained using AdamW", but then weight decay = 0.0, so isn’t this just vanilla Adam?
- For the CBS, do you get the numbers directly from Zhang, or do you use their power law estimates based on tokens, or something else?

**Questions:**

- Suppose I apply the Merrill et al approach at exactly those points where I drop the LR by 2x.  That is, I double the batch size, and then scale the LR back up by √2 (after dropping it by 2).  So the net change to the LR is to change by √2/2 = 1/√2.  Doesn’t this result in the exact prescription from this paper: whenever you would drop the LR by 2x, you instead double the batch size and decrease the LR by 1/√2?  If so, is the key difference from the Merrill approach just the timing of when you do this adjustment?  How does their proposed timing differ from yours?

- For Figure 3, what does “Seesaw” mean in terms of α and β?

---

> ### Author Response · Authors · 2025-11-19
> **response**
>
> We thank the reviewer for the thorough feedback on our work and we will address their concerns pointwise.
> ```
> The idea that Adam is basically normalized SGD, and that this necessitates dividing the learning rate by the square root of the batch size adjustment, was previously articulated in, e.g., in "Hilton - Batch size-invariance for policy optimization - 2110.00641v3", where (citing Hardin 2017), they similarly note "Adam divides the gradient by a running estimate of the root mean square gradient". Hilton et al. divide this estimate into the square root of {gradient-mean-squared plus gradient-variance}, and show how the batch size reduces the variance, and essentially leads to the square root LR adjustment. Since this paper wasn't discussed or cited, I'm not sure what the connections are, or the extent to which the theory as presented goes beyond this.
> ```
> We thank the reviewer for mentioning the paper by Hilton et al. and we apologize for missing it, and we will cite it in the final version of the paper. The crucial difference between the scaling rule proposed by Hilton et al. (and other subsequent follow-up works such as Malladi et al.) and our work is that their scaling only refers to training at a fixed constant batch size during training. Namely, fix batch size $B$ and tune learning rate $\eta$. Assuming training with Adam, their scaling rule says that if we now want to start our training with a fixed constant batch size $2B$, we have to divide the learning rate by $sqrt{2}$, without having to retune it. Our work however proposes a dynamical - during training - scheduler for the learning rate and batch size.
> ```
> Regarding soundness, I was concerned why we tested the extreme values of the equivalence (Figure 2) at 2x the CBS? Especially since they're much closer at 256, it makes me suspect that they're even closer at 128, yet this wasn't presented.
> ```
> In general, for 150M the CBS is roughly 256, hence the plot in Figure 2 is at CBS and slightly above. We have also now added Figure 9 in Appendix G with the plot at batch size 128. The reason for testing the extreme values is to show that our theory can predict the most aggressive schedule we can use (Lemma 3), and that indeed empirically this is confirmed. It is reasonable to expect that at smaller batch sizes it would take multiple increase steps in order to start running into the instabilities predicted by Lemma 3, hence why at 128 the losses are closer together. Nevertheless, the trend is present in the 3 plots as we increase the batch size.
>
> ```
> Also regarding soundness: I just feel there isn't the depth and breadth of experiments that we would normally find in an ICLR paper. Beyond this, I feel like there's not enough depth and breadth to convince me to use this approach in my own training.
> ```
> We apologize for not providing sufficient empirical backing for our work and we would like to kindly ask the reviewer what experiments they would expect to see and we will provide them during the rebuttal period.
>
> ```
> Limitations were not discussed, e.g., not acknowledging that state-of-the-art models are trained with non-zero weight decay, to higher tokens-per-parameter, etc. That this was only on one dataset with one tokenizer with one optimizer, etc.
> ```
> We now provide experiments with non-zero weight decay in Appendix E as well as 4x Chinchilla runs at 150M. We cannot however provide multiple pretraining datasets at multiple tokenizers due to compute constraints. We believe that in order to make scientific claims, we would need to do sweeps over learning rate, weight decay, datasets, tokenizers, which we cannot afford. Regarding optimizers, indeed our method is applicable to Adam(W) and other Adam-like variants such as SOAP and we are happy to make this more specific in the title and in the abstract of our work.
> ```
> The claim in the abstract that we "approach the theoretical limit implied by our analysis," is misleading, as approaching this limit is not, like, a sign the model is training well or whatever, but just that the continuous-time approximation that you used returns a number that is close to the actual number of steps, right? I mean, you could run a simple dry-run simulation to exactly determine how many steps the model will take given your algorithm, it's not like "abstract-level-ITALICIZED-claim" significant that your continuous-limit version provides a close answer, right? The significant thing is that the losses match, but, like I said, that's got nothing to do your continuous-time version. Unless I'm really misunderstanding something, which is possible, because...
> ```
> Indeed, our claim in the abstract is only under the condition that the losses match. Our claim in the abstract indeed states that ‘Seesaw matches cosine decay’. Here by matches we mean that the loss curves match. Also, our theory is for a proper discretized SGD process and not a continuous time approximation. We hope this clears the reviewer’s question.

---

> > ### Author Response · Authors · 2025-11-19
> > **part 2**
> >
> > ```
> > t's confusing as heck to use α and β to represent the adjustments to the LR and batch size --- used in the product that should remain constant --- and to use them as actual values that get substituted into this equation, essentially α := √α, β := (√α)^2. You know what I mean? Like, setting β = α is needed in order to satisfy α = √β. It's almost non-sensical.
> > "where the schedulers are equivalent in terms of loss as long as we keep the product α√β fixed" – what schedulers??? The one where B doesn’t change and the one where it does?
> > Let’s use α and β to be the adjustments to the LR and batch size, respectively.
> > Case 1: α = C and β = 1. I.e., the points where the LR schedule would drop by C and we don’t change the batch size. α√β = C * √1 = C. (e.g., C=2 would give us the abstract of the paper)
> > Case 2 (proposed): α = √C, and β = something. To maintain the invariant, β = C, only this way will α√β = √C√C = C as before.
> > As far as I can tell, that’s our only option for β in order to keep the product fixed. But then we say, of all the ways we could adjust the batch size, i.e., of all the β values, the most aggressive we can use must satisfy α = √β, i.e., β = α^2. So if α = √C, β = C. But isn’t this our only option to have an equivalent scheduler?
> > I REALLY wanted to understand what you’re saying here, but I just could not. Maybe on a second reading of the paper... but I'm just a reviewer, make my life easier please!
> > ```
> > We apologize for causing confusion with the current theorem, and we would like to provide a better statement. Firstly, we have updated the draft to contain the general version of the statement, for arbitrary decay factors. What our theorem says (for SGD, with a simple change for NSGD) is the following. Suppose we have one SGD process where we decay the learning rate by $\alpha_1$ and increase the batch size by $\beta_1$, and another SGD process where we decay the learning rate by $\alpha_2$ and increase the batch size by $\beta_2$, each process adjusted such that the total amount of data seen is the same. Then, if $\alpha_1 \beta_1 = \alpha_2 \beta_2$, we show that the risks of the 2 processes are within a constant factor of each other. As a corollary, under the variance dominated assumption, this result also holds for NSGD as long as $\alpha_1 \sqrt{\beta_1} = \alpha_2 \sqrt{\beta_2}$. We hope that this will clear up the confusion and we are happy to provide further explanations if necessary.
> >
> > ```
> > Not enough context prior to the experiments: E.g., there's a section on “Assumption 3”, but this assumption is only mentioned parenthetically before this point
> > I don’t really understand why all the content that was collected into Section 3 is in there:
> > Intuition
> > Theoretical results, with lots of pointers to other parts of the paper.
> > The formal algorithm
> > ```
> > Our goal has been to write the paper as first introducing Seesaw and the empirical benefits of our algorithm, followed by the theoretical derivation. For exposition, we were aiming to have an informal view of the variance dominated assumption in Section 3.1, followed by formalizing it later in Assumption 3 in the theory section, Section 5. We will change the exposition in the final version of our paper.
> >
> > ```
> > Like, you say, “the NSGD update rule, which is a crucial component of designing Seesaw”, but it seems to me that using a simple square-root scaling rule would have been sufficient here.
> > ```
> > We would like to point out that the “square-root scaling rule” can have 2 separate connotations. As we mentioned in the previous point, previous literature has studied the square-root scaling rule in the context of a fixed batch size training run, not a ramp-up during training. We study a batch size scheduler, derived based on the analysis of NSGD on quadratics, which gives rise to our algorithm.

---

> > > ### Author Response · Authors · 2025-11-19
> > > **part 3**
> > >
> > > ```
> > > When you present your main findings (Table 1), maybe provide some interpretation? Like, what is the take-home message here?
> > > ```
> > > Table 1 contains the final losses from Figure 1, which we have written down for easier comparison based on the models. We have mentioned in the caption that the dynamics of the 2 schedulers (meaning the final losses, as well as the actual loss curves during training as one can see from Figure 1) match robustly between Seesaw and Cosine when trained at CBS. We are happy to provide further explanations if necessary.
> > > ```
> > > Too much required for understanding is in the appendices.
> > > ```
> > > Due to the page limit, we were forced to put the proofs for our theorems and the main analysis in the Appendices, while leaving only the main statements themselves in the manuscript, with intuitive explanations provided in Section 3. We believe this is standard practice when page limit is a constraint, but we are happy to include additional information in the main body if the reviewer could point us to what they consider would be useful for understanding.
> > > We appreciate the reviewer pointing out the typos and we have fixed them. Moreover we will provide a more detailed Taylor expansion derivation in the final version of our paper. For the CBS estimates, we use the estimates from Zhang et al.
> > >
> > > ```
> > > Suppose I apply the Merrill et al approach at exactly those points where I drop the LR by 2x. That is, I double the batch size, and then scale the LR back up by √2 (after dropping it by 2). So the net change to the LR is to change by √2/2 = 1/√2. Doesn’t this result in the exact prescription from this paper: whenever you would drop the LR by 2x, you instead double the batch size and decrease the LR by 1/√2? If so, is the key difference from the Merrill approach just the timing of when you do this adjustment? How does their proposed timing differ from yours?
> > > ```
> > > Yes, the reviewer is technically correct. However, we think this is philosophically a different argument as proposed in Merrill et al. To the best of our understanding, Merrill et al. proposed to follow the base schedule, while trying to increase batch size by 2 and lr by sqrt(2) at various points and checking if the network performs well for a short time schedule. Instead, our method proposes to change the underlying learning rate and batch size schedule in tandem right from the start to keep the underlying process the same as a fixed batch size process. For implementation purposes, we need to discretize the underlying learning rate schedule so that we don’t have to modify the batch size many times in a single run. The discretization is an artefact to help the method be implementable in practice and is not directly connected with the underlying idea of our work.
> > > ```
> > > For Figure 3, what does “Seesaw” mean in terms of α and β?
> > > ```
> > > Figure 3 is under the same setup as Figure 1, i.e. $\alpha=\sqrt{1.1}$ and $\beta=1.1$.

---

> ### Comment · Reviewer_gioo · 2025-11-27
>
> I'd like to thank the authors for their detailed response. I definitely intended my initial review to be constructive, i.e., "here is what a motivated reviewer struggled with during a reading of your paper and what you can do to improve," and not so much, "please explain to me in your rebuttal the points that I was confused about."  I see you provided a revised paper without detailed changes, but I will assume that you will make the changes that you said you would make for the final camera copy.  I also think you should think hard about all my other comments, they really are for your benefit here, and make changes there too.  I really think all my criticisms stand, e.g., you should specifically state and acknowledge limitations (note in that comment I wasn't criticizing your paper for HAVING limitations---there are always limitations, it's perfectly fine to have limitations---it's just very helpful for readers to know what they are), you should think harder about revising the Table 1 caption (note you don't say in Section 4 or in the caption itself that these are final losses for the curves in Figure 1, the reader would have to remember from Section 1 that you had a forward-pointer to Table 1, and then figure out what you mean by "training dynamics", and then try to align which curves in Figure 1 correspond to the CBS, and then remember why this is important to note).
>
> I note that you explicitly acknowledge "not providing sufficient empirical backing for our work" -- I am going to assume that this was a passive-aggressive "non apology" and that the authors do not actually feel, upon reflection, that their work does not have sufficient empirical backing.  But definitely correct me if I'm wrong here!
>
> Regarding the weight decay experiments:
> - Thank you for adding those. As I understand it, the take-home is "decoupled weight decay does not affect our scheme".  I'm actually fairly surprised that the optimal LR and WD didn't change as the batch size increased 4x.  I would think you would also be surprised about this, given your main contribution involves making changes to the LR as the batch size changes (albeit dynamically).  Might be worth adding some discussion of this to that section.
> - if the optimal LR is not sensitive to BS changes, it makes me wonder if you should do an experiment where you keep the same LR schedule, but change the batch size in the same places that Seesaw would adjust it.  How does that perform?  What can we learn from it?
>
> Assuming the authors will improve their presentation, and given they've added some additional experiments, I think the paper definitely moves from a clear "reject" to a "borderline reject, but I wouldn't be upset if it got in".  That captures how I feel about it fairly well at this stage, being very optimistic about the revisions and with a lot of good will.

---

### Official Review · Reviewer_nmrR · 2025-10-28

**Soundness:** 2
**Presentation:** 2
**Contribution:** 3
**Rating:** 6
**Confidence:** 3

**Summary:**

This paper proposes Seesaw, a principled batch size scheduling algorithm designed to reduce wall-clock training time for LLM pre-training. The key insight is to replace LR decay with slower decay and increasing batch size. Specifically, when a standard schedule would decrease LR by factor $\alpha$, Seesaw instead decays LR by $\sqrt{\alpha}$ while increasing batch size by $\alpha$.

The authors motivate this rule through theoretical analysis of SGD on noisy linear regression, extending to Normalized SGD (NSGD) as a proxy for Adam. Under a "variance-dominated" regime, analysis on NSGD results in the Seesaw scheduler. Experiments on 150M-600M parameter models at Chinchilla scale demonstrate ~36% wall-clock time reduction while matching baseline validation loss.

**Strengths:**

1. **Strong empirical results.** The paper's main contribution is its compelling empirical validation: Seesaw achieves approximately 36% reduction in wall-clock training time while matching the baseline's final validation loss. For practitioners with access to parallel compute resources (for processing larger batch sizes), this translates directly to significant cost savings. Another advantage is its simplicity, as it can be implemented as a straightforward drop-in replacement for standard learning rate schedules.
2. **Principled derivation of a simple heuristic.**  The authors bridge theory and practice by grounding their scheduler in a principled theoretical framework rather than relying on pure empirical tuning. This approach elevates batch size scheduling from ad-hoc experimentation to a more systematic methodology.
3. **Rigorous validation of theoretical predictions and failure modes.** The evaluation of where its theory succeeds and where it breaks down strengthens the paper. Figure 2 validates the stability condition derived in Lemma 3, confirming that overly aggressive schedules ($\alpha < \sqrt{\beta}$​) lead to performance degradation as predicted. Figure 3 demonstrates the failure of the "variance-dominated" assumption at very large batch sizes, where Seesaw no longer tracks the baseline. This transparency about limitations adds credibility and helps practitioners understand the method's applicability boundaries.

**Weaknesses:**

1. **The NSGD proxy inadequately represents Adam.** The choice of Normalized SGD appears driven by mathematical tractability rather than fidelity to Adam's behavior. NSGD's global L2 normalization differs fundamentally from Adam's coordinate-wise adaptivity. A strong consensus in recent literature suggests that SignSGD, which respects the sign-based nature of Adam's updates, is a much better conceptual proxy. Since Seesaw's square-root scaling rule is specifically tailored to the analysis on NSGD, it's unclear whether similar insights hold for SignSGD or Adam. The paper should explicitly acknowledge this limitation and discuss why the NSGD-derived heuristic succeeds despite this mismatch.

2. **Oversimplified theoretical foundation.** The noisy linear regression setting vastly oversimplifies LLM training's non-convex, high-dimensional landscapes. More critically, the "variance-dominated" assumption (Assumption 3) appears overly strong and may only hold near the end of training. The paper does not sufficiently justify why analyzing this simplified setting provides a good proxy for LLM training dynamics, even though the empirical results suggest the heuristic works in practice.

3. **Performance degradation at large batch sizes.** While overall results are strong, Table 1 shows Seesaw slightly underperforming the cosine baseline at large batch sizes (e.g., $B=1024$ for all 150M, 300M, 600M models). This gap along with Figure 3 suggest diminishing benefits as batch size grows beyond the variance-dominated regime. This ceiling on applicability could limit the method's utility for larger-scale training runs in practice.

4. **Proof presentation and limited technical novelty.** The proofs in Appendix A suffer from unclear notation. Multiple symbols (e.g. $Q$, $\lambda$, $\Lambda$) are used without proper definition, making the derivations difficult to follow. Presumably these relate to an eigendecomposition $H=Q\Lambda Q^T$ with $\lambda = \text{diag}(\Lambda)$, but this is never explicitly stated. Additionally, the theory specifically analyzes a factor of 2.0 step decay but does not provide a proof for general drops by factor $\alpha$, limiting its generality. Beyond these presentation issues, the technical contribution itself is limited—the proof is straightforward and follows the same approach as Meterez et al. (2025).

5. **Inconsistent visualization in Figure 1.** The figure uses log scale for tokens but linear scale for steps, which is inconsistent and potentially misleading. The log scale can obscure small but meaningful differences in sample efficiency, while the linear scale visually exaggerates the speedup. Using consistent linear scaling for both plots would provide a more transparent and fair comparison of the method's performance.

**Questions:**

1. Could the authors elaborate on why NSGD was chosen as a proxy for Adam, given that recent literature increasingly favors SignSGD as a more faithful conceptual model? Would a theoretical analysis based on SignSGD yield the same $\sqrt{\alpha}$​ scaling rule, or would different dynamics emerge?

2. The theory analyzes the specific case of a factor 2.0 decay. Could the authors provide a proof for the general case of decay by factor $\alpha$?

3. The practical implementation discretizes the continuous cosine decay (e.g., using $\alpha=1.1$ in Table 1). How sensitive is the method's performance to this discretization factor? Would finer-grained approximations better track the cosine curve, and are there practical trade-offs to consider in choosing $\alpha$?

4. Could the authors revise Figure 1 to use consistent linear scaling for tokens to enable fairer visual comparison?

---

> ### Author Response · Authors · 2025-11-19
> **response**
>
> We thank the reviewer for their kind words regarding our method and paper, and we will proceed by addressing the weaknesses bulletwise.
>
> ```
> The NSGD proxy inadequately represents Adam. The choice of Normalized SGD appears driven by mathematical tractability rather than fidelity to Adam's behavior. NSGD's global L2 normalization differs fundamentally from Adam's coordinate-wise adaptivity. A strong consensus in recent literature suggests that SignSGD, which respects the sign-based nature of Adam's updates, is a much better conceptual proxy. Since Seesaw's square-root scaling rule is specifically tailored to the analysis on NSGD, it's unclear whether similar insights hold for SignSGD or Adam. The paper should explicitly acknowledge this limitation and discuss why the NSGD-derived heuristic succeeds despite this mismatch.
> ```
> We will add a section explaining the limitation of our analytical model, and the comparison with SignSGD. Indeed, in order to have a tractable analytical model for Adam, we have to remove certain components from the whole optimizer. While signSGD is a better model than NSGD in terms of per-coordinate adaptivity, the simpler proxy NSGD does capture the mean and variance components in the Adam denominator which turned out to be the crucial components for deriving Seesaw. Moreover, previous works have shown empirically that NSGD (in [1] termed Adalayer) is a sensible proxy to use for Adam in terms of performance. Furthermore, the NSGD proxy has high predictive power in our analysis, as it not only shows when Seesaw will work, but it also predicts the failure modes of our scheme, as the reviewer also kindly pointed out in the strengths section.
>
> ```
> Oversimplified theoretical foundation. The noisy linear regression setting vastly oversimplifies LLM training's non-convex, high-dimensional landscapes. More critically, the "variance-dominated" assumption (Assumption 3) appears overly strong and may only hold near the end of training. The paper does not sufficiently justify why analyzing this simplified setting provides a good proxy for LLM training dynamics, even though the empirical results suggest the heuristic works in practice.
> ```
> While the noisy linear regression model is indeed an oversimplification of the LLM landscape, as we can see in our experiments it does have a strong predictive power, and we would argue that its simplicity is a strength of the model, and not a weakness. In order to support our method further, we run experiments at 4X Chinchilla for 150M models in Appendix F at batch sizes 256, 512, 1024, and we show that even in this regime Seesaw matches cosine, making our method practically useful in realistic training scenarios.
>
> ```
> Performance degradation at large batch sizes. While overall results are strong, Table 1 shows Seesaw slightly underperforming the cosine baseline at large batch sizes (e.g.,  for all 150M, 300M, 600M models). This gap along with Figure 3 suggest diminishing benefits as batch size grows beyond the variance-dominated regime. This ceiling on applicability could limit the method's utility for larger-scale training runs in practice.
> ```
> While the performance degradation does indeed happen once crossing CBS (CBS values mentioned in lines 258-262), note that training with a batch size past CBS is not a practical training regime in general and thus does not reduce the usefulness of Seesaw. However, as argued in the previous point, the variance dominated assumption could indeed fail at initialization for very long training runs where the CBS is much larger - but note however that these are 2 separate points.

---

> > ### Author Response · Authors · 2025-11-19
> > **part 2**
> >
> > ```
> > Proof presentation and limited technical novelty. The proofs in Appendix A suffer from unclear notation. Multiple symbols (e.g. , , ) are used without proper definition, making the derivations difficult to follow. Presumably these relate to an eigendecomposition  with , but this is never explicitly stated. Additionally, the theory specifically analyzes a factor of 2.0 step decay but does not provide a proof for general drops by factor , limiting its generality. Beyond these presentation issues, the technical contribution itself is limited—the proof is straightforward and follows the same approach as Meterez et al. (2025).
> > ```
> > We thank the reviewer for pointing out the missing notation and we have added it in the manuscript in Appendix A.1. Moreover, we have now added the general statement and proof for arbitrary $\alpha$ and $\beta$, by rewriting Theorem 1 and Corollary 1, as well as adding the proof for this new statement in Appendix A. However, we would like to gently push back on the claim that our technical contribution is limited. While our proof approach does follow Meterez et al. (2025) as it is the standard treatment for quadratic analysis, the claim is completely different and not covered in previous literature. Meterez et al. study SGD with weight averaging and a constant learning rate and constant batch size providing an **upper bound on the excess risk**. In contrast, we study SGD with a learning rate schedule and batch size schedule, and prove (under our step schedulers) that the exact risk of an SGD process with fixed learning rate and changing batch size, and an SGD process with changing learning rate at fixed batch size, are within a constant factor of each other - a claim which has not been previously shown in literature.
> >
> > ```
> > Inconsistent visualization in Figure 1. The figure uses log scale for tokens but linear scale for steps, which is inconsistent and potentially misleading. The log scale can obscure small but meaningful differences in sample efficiency, while the linear scale visually exaggerates the speedup. Using consistent linear scaling for both plots would provide a more transparent and fair comparison of the method's performance.
> > ```
> > It is not our intent to be misleading, but the opposite: logscale amplifies small differences on a plot, and thus provides more detail as opposed to hiding meaningful differences. We have now provided Figure 1 in linear-linear scale, as Figure 10 in Appendix G. Note that we do not claim Seesaw achieves better sample efficiency - we claim that for the same amount of tokens (samples), Seesaw achieves a serial step reduction. Hence, plotting with tokens on the x axis we expect the losses to overlap, whereas plotting with steps on x axis we expect a reduction.
> >
> > ```
> > Could the authors elaborate on why NSGD was chosen as a proxy for Adam, given that recent literature increasingly favors SignSGD as a more faithful conceptual model? Would a theoretical analysis based on SignSGD yield the same ​ scaling rule, or would different dynamics emerge?
> > ```
> > As previously mentioned, NSGD was chosen due to its analytical tractability and predictive empirical power. While in 1D, NSGD is roughly similar to SignSGD (up to taking expectation of the gradient norm in the denominator), we cannot speculate as to the dynamics of SignSGD under our scheme and believe this is an interesting avenue for future work.
> >
> > ```
> > The theory analyzes the specific case of a factor 2.0 decay. Could the authors provide a proof for the general case of decay by factor ?
> > ```
> > We have now added the general statements and analysis for arbitrary factors in Theorem 1, Corollary 1 and Appendix A.
> >
> > ```
> > The practical implementation discretizes the continuous cosine decay (e.g., using  in Table 1). How sensitive is the method's performance to this discretization factor? Would finer-grained approximations better track the cosine curve, and are there practical trade-offs to consider in choosing ?
> > ```
> > Yes, we do think the method depends on the discretization factor being used. However, in our experiments, we simply picked a discretization factor of 1.1 and the loss curves almost match. We would assume we could have picked a slightly higher discretization factor without any degradation in performance.
> >
> > ```
> > Could the authors revise Figure 1 to use consistent linear scaling for tokens to enable fairer visual comparison?
> > ```
> > We have provided Figure 1 in lin-lin scale as Figure 10 in Appendix G.
> >
> > [1] Zhao, Rosie, Depen Morwani, David Brandfonbrener, Nikhil Vyas, and Sham Kakade. "Deconstructing what makes a good optimizer for language models." arXiv preprint arXiv:2407.07972 (2024).

---

> > > ### Comment · Reviewer_nmrR · 2025-11-28
> > >
> > > Thank you for the detailed response. I particularly appreciate your new general statement and proof for arbitrary α and β in Theorem 1 and Corollary 1.
> > >
> > > My remaining concern is the choice of NSGD as a proxy for analyzing Adam. The authors justify this by noting that: (1) NSGD (termed "Adalayer" in [1]) achieves similar performance to Adam empirically, (2) NSGD resembles SignSGD in 1D, and (3) the theory shows good predictive power. I find these justifications insufficient:
> > >
> > > (1) The Adalayer comparison is misleading. Adalayer in [1] uses layer-wise adaptive learning rates, fundamentally different from NSGD's uniform normalization. This layer-wise adaptation was likely crucial to matching Adam's performance.
> > >
> > > (2) The 1D analogy doesn't generalize. NSGD and SignSGD can behave drastically differently in high dimensions, which are more relevant to practice.
> > >
> > > (3) Predictive power doesn't justify the modeling choice. While experiments match theory well, there are two major gaps: objective (noisy linear regression vs. LLM pretraining) and optimizer (NSGD vs. AdamW). Simplifying the objective is standard practice for developing theory, but the optimizer choice directly affects your main theoretical contribution—the optimal batch size schedule. This makes NSGD a problematic proxy.
> > >
> > > My recommendation is to add a Limitations section explicitly discussing the NSGD choice and its implications. I cannot find such discussion in the current manuscript.
> > >
> > > I will maintain my positive score as the theoretical contributions and experimental validation are valuable, but I cannot increase it further given remaining concerns.

---

### Official Review · Reviewer_fg72 · 2025-10-31

**Soundness:** 2
**Presentation:** 3
**Contribution:** 3
**Rating:** 4
**Confidence:** 4

**Summary:**

The proposed SEESAW scheduling method replaces standard learning rate halving by concurrently multiplying the learning rate by \sqrt{\frac{1}{2}}  and doubling the batch size. This reduces serial steps while preserving the loss trajectory. Experiments on 150M-600M models with C4 and Chinchilla scaling show SEESAW matches cosine scheduling's final loss while achieving near-theoretical-maximum speedups.

**Strengths:**

1. The paper establishes a non-asymptotic equivalence between learning rate decay and batch size growth under SGD, extending it to NSGD via an equivalence family where the product \alpha \sqrt{\beta} is conserved. This links theoretical insight to practice, forming an actionable framework for designing training protocols.

2. The proposed algorithm features a remarkably simple structure and achieves true zero intrusion, meaning it can be seamlessly integrated into existing training pipelines without requiring any modifications to the model architecture, optimizer, or other components. Its plug-and-play nature makes it highly accessible and easy to adopt in practice.

3. The theoretical upper bound derived in the paper aligns closely with experimental results.

**Weaknesses:**

1. The theoretical derivation relies heavily on Assumption 3; however, Figure 3 shows that as the batch size increases, Seesaw diverges from cosine scheduling, indicating a fundamental limitation in the method's applicability when this assumption breaks down.

2. The paper lacks comprehensive experimental validation across a broader range of conditions. It evaluates only three medium-scale models on a single dataset and omits comparisons across diverse downstream tasks, additional optimizers, or extensive hyperparameter settings. More extensive experiments would be necessary to convincingly demonstrate the generalization and robustness of the proposed method.

3. The notion of acceleration is measured in terms of reduced serial steps rather than actual wall-clock time. While the paper reports the ratio of achieved serial step reduction to the theoretical limit, it does not provide end-to-end wall-clock evaluations under realistic distributed training scenarios, including cross-node throughput, communication overhead, and memory constraints. Furthermore, extreme members of the proposed equivalence family exhibit instability in practice, raising concerns about their usability.

**Questions:**

Please refer to the weaknesses

---

> ### Author Response · Authors · 2025-11-19
> **response**
>
> We appreciate the reviewer’s feedback and we will address their weakness claims pointwise.
> ```
> The theoretical derivation relies heavily on Assumption 3; however, Figure 3 shows that as the batch size increases, Seesaw diverges from cosine scheduling, indicating a fundamental limitation in the method's applicability when this assumption breaks down.
> ```
> Indeed, note that Seesaw only works in the variance dominated regime. In our experiments, this assumption seems to break down when we are past the critical batch size (values mentioned in line 258-262). However, from a practical point of view, we argue that this is not an issue since generally we do not train models in this regime: training past the critical batch size is wasteful in terms of number of samples. Thus while this is a fundamental limitation, it does not limit the applicability of Seesaw in any practical training regime.
> ```
> The paper lacks comprehensive experimental validation across a broader range of conditions. It evaluates only three medium-scale models on a single dataset and omits comparisons across diverse downstream tasks, additional optimizers, or extensive hyperparameter settings. More extensive experiments would be necessary to convincingly demonstrate the generalization and robustness of the proposed method.
> ```
> We now provide weight decay experiments in Appendix E as well as downstream eval comparisons on 150M for PIQA, HellaSwag, and ARC Easy in Figure 11, Appendix G. Note that for all our runs we have run extensive hyperparameter sweeps, as detailed in Section 4, with the weight decay sweep detailed in Appendix E. With respect to optimizers, our method is applicable to Adam(W), as well as SOAP (which is Adam in a rotated basis). It would be an interesting future direction to rederive the theory for other optimizers such as Muon or Shampoo. We are happy to change both the title and abstract to reflect the fact that our method is only applicable to Adam (and Adam-like optimizers).
> ```
> The notion of acceleration is measured in terms of reduced serial steps rather than actual wall-clock time. While the paper reports the ratio of achieved serial step reduction to the theoretical limit, it does not provide end-to-end wall-clock evaluations under realistic distributed training scenarios, including cross-node throughput, communication overhead, and memory constraints.
> ```
> We agree with the reviewer that wall clock time is an important consideration and that serial step reduction does not map directly onto wall clock reduction. Seesaw does reduce wall clock time under the assumption that for the increased batch size, the time per step does not increase. While in this work we do not take into account the communication overhead introduced by allocating more GPUs to a training run, we believe that these constraints are more on the engineering side and are unavoidable to any batch size scheduling scheme and can be addressed down the line. Moreover, recent frontier LLMs such as Llama and Apertus do indeed apply a batch size ramp up scheme, further confirming that this is a fruitful avenue of research.
> ```
> Furthermore, extreme members of the proposed equivalence family exhibit instability in practice, raising concerns about their usability.
> ```
> Is the reviewer referring to Figure 2 in Section 4.1? In this case, in the paper we use and suggest using $(\alpha, \sqrt{\alpha})$ for the scheduling rule as this is the maximal stable rule one can use as argued by Lemma 3. Indeed, the extreme members exhibit instability, which is line with our theoretical and empirical observations.

---

### Official Review · Reviewer_r7Qn · 2025-10-31

**Soundness:** 4
**Presentation:** 4
**Contribution:** 3
**Rating:** 8
**Confidence:** 2

**Summary:**

The authors present a theoretically grounded approach to batch size scheduling, specifically showing that decay schedules can be replaced by a combination of decay schedules and batch size schedules (e.g. instead of decaying by a factor of 2 you can scale LR by 2 and decay by a factor of sqrt(2)). This allows you to either exploit more hardware or (more realistically) exploit the hardware you have more effectively.

They conduct small experiments up to 600M "Chinchilla" that demonstrate that the theory works out, showing both that their schedule is equivalent and also that a more aggressive schedule (in terms of higher effective LR) won't work, or works less well.

**Strengths:**

Overall I think this is a nice paper: It is generally very well written. It's a nice example of trying to deploy actual theoretical insights with practical advice, and I think it's a fairly creative way of going about it. I think one could have gotten to the same conclusions from older SDE theory about Adam LR's relationship to batch size, but it's good to have this new approach too.

a nice extension of the theory from Meterez, et al 2025 (and others) coupled with some small but very convincing experiments demonstrating the idea.

The experiments show that their theory holds up remarkably well.

This isn't a world-shattering paper but, modulo my concerns, it is a solid contribution and one i'd be keen to test myself.

**Weaknesses:**

It is easy to say that it would be nice to see bigger experiments, but that doesn't feel necessary here.

One small thing is that the authors appear to have moved their theoretical results to the end of the paper late in the drafting process: assumption 3 is referenced in 3.1 and 4.2 but not defined until section 5. This is confusing but easily remedied.

I think it's a little much to claim a 36% "wall clock" speedup since it would need ~proportionally more compute for those phases (or considerably smaller wall clock gains from improved MFU)... I get the point that it's possible to use more compute in those circumstances when you might otherwise be constrained by CBS (though there are other ways to do that too through model parallelism)

I am quite confused by how Lemma 3 is a refutation of the approach proposed by Merrill et al... They're trying to hold effective LR constant. They increase B by \beta = 2, which would reduce the effective LR by sqrt(2) (consistent with your analysis of NSGD and other prior results with SDE etc.) But they then increase the LR by a factor sqrt(2), and so alpha=sqrt(2), and so the effective LR is held constant? Right?

The absence of weight decay in the experiments makes sense given its weird interaction with LR, but it's not realistic. In particular, decoupled weight decay means LR impacts steady state of the weight norms, and so you would end up with different results... Does the theoretical analysis transfer to this more realistic setting?

**Questions:**

Clarification about Lemma 3's relationship to Merrill et al 2025 would be helpful. I feel like I'm missing something.

some experiments with actual weight decay would be helpful since that is how LLMs are usually trained.

I'm also curious about behavior at sub-CBS (e.g. 150M with B=128).

Is it possible to exploit the empirical observation from Merrill et al 2025 that their measured CBS seems to increase as training progresses? Or is your contention that their estimate of CBS is inherently flawed?

relatedly, it would be nice to see this holding for the overtrained ( say 4x Chinchilla) regime too. I wouldn't expect to see all scales to a multiple of chinchilla. Given that many real models are overtrained, it would be nice to see. (This may be asking too much and that's ok.)

---

> ### Author Response · Authors · 2025-11-19
> **response**
>
> We thank the reviewer for their kind words regarding our work and we will address their concerns pointwise.
> ```
> One small thing is that the authors appear to have moved their theoretical results to the end of the paper late in the drafting process: assumption 3 is referenced in 3.1 and 4.2 but not defined until section 5. This is confusing but easily remedied.
> ```
> We acknowledge this mistake on our end. We will (informally, for the sake of exposition, tying into the introduction of section 3.1) refer to this assumption is sections 3 and 4 as the “variance dominated assumption”, and point to a formalized version of it in Assumption 3.
>
> ```
> I think it's a little much to claim a 36% "wall clock" speedup since it would need ~proportionally more compute for those phases (or considerably smaller wall clock gains from improved MFU)... I get the point that it's possible to use more compute in those circumstances when you might otherwise be constrained by CBS (though there are other ways to do that too through model parallelism)
> ```
> Indeed, our speed-up is in terms of serial steps. The wall-clock speed-up is under the assumption that with the increased batch size, the time per step would stay constant. This is possible (ignoring distributed systems constraints such as communication etc., which we acknowledge pose a significant challenge in practice) by allocating multiple GPUs and increasing the degree of parallelism, which we believe is similar to the regime frontier labs operate in.
>
> ```
> I am quite confused by how Lemma 3 is a refutation of the approach proposed by Merrill et al... They're trying to hold effective LR constant. They increase B by \beta = 2, which would reduce the effective LR by sqrt(2) (consistent with your analysis of NSGD and other prior results with SDE etc.) But they then increase the LR by a factor sqrt(2), and so alpha=sqrt(2), and so the effective LR is held constant? Right?
> ```
> Note that in our scheme batch size is increased by $\beta$, and learning rate is decreased by $\alpha$, thus the scheme proposed by Merril et al. sets (in our notation), $\beta = 2$ and $\alpha = 1/\sqrt{2}$, which, by Lemma 3 and the empirical ablations in Section 4.1, would lead to blow up after a large enough number of steps.
>
> ```
> The absence of weight decay in the experiments makes sense given its weird interaction with LR, but it's not realistic. In particular, decoupled weight decay means LR impacts steady state of the weight norms, and so you would end up with different results... Does the theoretical analysis transfer to this more realistic setting?
> ```
> We have now provided empirical results showing that decoupled weight decay does not affect our scheme in Appendix E.
>
> ```
> I'm also curious about behavior at sub-CBS (e.g. 150M with B=128).
> ```
> Note that we already provide 150M at B=128 in Table 1. Moreover, we also provide sub-CBS results in the same table due to the following argument. Following Zhang et al, we know that the CBS (in case of Chinchilla scaling, i.e. data size $D$ proportional to parameter size $N$), as $\sqrt{D}$. Therefore, the CBS for 300M and 600M is larger than 256, but we do provide smaller batch size results in Table 1.
>
> ```
> relatedly, it would be nice to see this holding for the overtrained ( say 4x Chinchilla) regime too. I wouldn't expect to see all scales to a multiple of chinchilla. Given that many real models are overtrained, it would be nice to see. (This may be asking too much and that's ok.)
> ```
> We have now added experiments on $4 \times$ Chinchilla for 150M models in Appendix F, showing that even in this regime Seesaw matches cosine decay.

---

### Author Response · Authors · 2025-11-19
**General answer to the reviewers**

We would like to thank the reviewers for their thorough feedback. We have addressed bulletwise the concerns of each reviewer, and we will use this general statement to mention the main changes provided to the manuscript. We have colored the main changes in the manuscript in blue.

In terms of changes, both experimental and theoretical, we now provide:
- Experiments showing that for a 150M model trained at 4x Chinchilla, across 256, 512 and 1024 (at sequence length 1024) batch sizes, Seesaw and cosine still have matching final losses
- Weight decay experiments for 150M in Appendix E
- Downstream eval results for the 150M models on PIQA, HellaSwag and ARC Easy in Figure 11, Appendix G, showing that cosine and Seesaw achieve similar downstream performance across multiple seeds
- In Section 5, we have now added the generalized Theorem 1 and Corollary 1 for arbitrary decays (instead of just halving/doubling), as well as the corresponding proof in Appendix A

---

### Comment · Area_Chair_wmMw · 2025-11-28

Dear Reviewers,

The discussion phase is now underway, and the authors have finished uploading their responses to reviewers. If you haven't already, please carefully review the authors' responses to understand their perspectives. Engage in thoughtful, constructive discussions with authors, sharing your thoughts and seeking clarifications. Please also update your review or rating if necessary.

It is noted in the guideline that reviewers can leave comments visible to authors **until Dec 2 11:59pm AoE**. Your active participation and contribution to the ongoing discussion are highly encouraged. Thank you very much for your contribution to ICLR.

Best regards,

AC

---

### Meta-Review · Area_Chair_sqtB · 2026-01-06

**Summary:**

Across reviews, the paper was generally appreciated as a simple, practical recipe for trading learning-rate decay for batch-size growth, supported by a clean theoretical framing and empirical validation on 150M–600M-scale pretraining runs. The main concerns were about (1) generality, (2) clarity of claims/metrics, and (3) faithfulness of the theoretical proxy to practice. The rebuttal adds meaningful empirical and theoretical strengthening, but not all concerns were addressed, e.g., NSGD-proxy/limitations discussion, experimental “depth and breadth”.

**Reviewer Concerns:**

**Reviewer r7Qn**

Addressed

Confusing placement of Assumption 3: authors acknowledge and propose clearer signposting.

“36% wall clock” phrasing: clarified as serial-step speedup under assumptions about parallelism.

Lemma 3 vs Merrill et al.: authors explain notation mapping and why Merrill-style scaling leads to blow-up per Lemma 3, plus empirical support.

Weight decay realism: added weight decay experiments.

Overtrained regime (4× Chinchilla): added experiments for 150M.

**Reviewer fg72**

Addressed

Dependence on Assumption 3 / divergence past CBS: method intended for variance-dominated regime; argues training past CBS is wasteful; limitation acknowledged.

Lack of broader experimental coverage: added weight decay and downstream tasks; clarified extensive sweeps and applicability to Adam-like optimizers.

Wall-clock vs serial steps: explains assumptions and notes engineering constraints.

Instability of extreme family members: clarified that the paper recommends the maximal stable rule, consistent with Lemma 3; extreme cases are expected to fail.

**Reviewer nmrR**

Addressed

Requested general proof beyond factor-2 decay: added generalized Theorem/Corollary and proof.

Proof notation and figure scaling concerns: add missing notation and provide linear-linear figure in appendix.

Weight decay / realism: added weight decay experiments; argues practical utility in realistic regimes.

Still outstanding

NSGD as proxy for Adam: will add a Limitations section discussing this gap and implications.

**Reviewer gioo**

Addressed (potentially)

Missing weight decay / limitations: weight decay experiments added; authors say they will add limitations and improve presentation.

Broader empirical backing: some additional experiments (weight decay, 4× Chinchilla; mention of additional plot at B=128; downstream tasks were added).

Prior work citation (Hilton et al.): authors acknowledge and will cite; they argue novelty is dynamic during training scheduling vs fixed-batch scaling rules.

Confusing theorem/notation/exposition: authors provide a clearer theorem statement and promise restructuring.

Still outstanding

Reviewer still believes many clarity/organization issues remain and expects substantial camera-ready editing (limitations section, Table 1 caption clarity, better interpretation, less reliance on appendices).

Reviewer still doubts overall experimental “depth and breadth” relative to ICLR norms, even after additions.

**Reviewer Scores:**

r7Qn (initial 8): potentially keep the score, rebuttal resolves their minor concerns.

fg72 (initial 4): May increase to 6. Their concerns were largely about experimental breadth and wall-clock framing; rebuttal adds key missing experiments and clarifies limitations. But absence of true distributed wall-clock likely keeps them cautious.

nmrR (initial 6): No change. They will not raise further due to unresolved NSGD-proxy/limitations discussion.

gioo (initial 2): Increase to 4. Reviewer signals a clear upward revision from “clear reject” to “borderline reject”. Not all concerns addressed.

---

### Decision · Program_Chairs · 2026-01-26

Accept (Poster)